# Marine particles and their remineralization buffer future ocean biogeochemistry response to climate warming

Joeran Maerz<sup>1,2</sup>, Katharina D. Six<sup>1</sup>, Soeren Ahmerkamp<sup>3,4</sup>, and Tatiana Ilyina<sup>1,5</sup>

**Correspondence:** Joeran Maerz (joeran.maerz@mpimet.mpg.de)

Abstract. Transport and fate of particulate organic carbon (POC) and nutrients through marine particles co-determine the future response of ocean biogeochemistry and oceanic carbon uptake under climate warming. This makes the parametrization of the biological carbon pump in Earth system models (ESMs) an important model component and motivates us to compare a recently developed new sinking scheme (M<sup>4</sup>AGO; Maerz et al., 2020) to the current CMIP6 default Martin curve-like sinking scheme in MPI-ESM1.2-LR (see Mauritsen et al., 2019) under the future shared socio-economic pathway high-emission scenario SSP5-8.5. In their global response, the two model versions are similar, showing a decrease of integrated net primary production between the historical (1985-2014) and future (2070-2099) period of about 8.1 \% and 9.7 \% for the CMIP6 and M<sup>4</sup>AGO version, respectively. However, the models response differs latitudinally. In M<sup>4</sup>AGO, the temperature-dependent remineralization offsets the future increase in sinking velocity caused by a higher CaCO<sub>3</sub> to POC ratio in the low latitudes. There, M<sup>4</sup>AGO thus buffers the export loss of nutrients to the mesopelagic, visible in little future changes of the export to net primary production ratio (the p ratio), while the CMIP6 version shows more pronounced changes with regionally declining or increasing p ratio. In the Arctic Ocean, the projected future increase of net primary production in the CMIP6 version is diminished with M<sup>4</sup>AGO through its higher POC transfer efficiency in high latitude regions. Hence, the more mechanistic and to environmental changes-responding M<sup>4</sup>AGO scheme shows a stronger buffering regional response to climate warming than the CMIP6 model version. The higher transfer efficiency also impinges on higher CO<sub>2</sub> uptake in high latitude regions while the tropical regions turn later into a net sink with M<sup>4</sup>AGO compared to the standard CMIP6 version. Next to ballasting, we identified the particle microstructure as vigorous determinant for future changes of POC sinking velocity. Microstructure co-determines particle porosity and particle density. Processes governing the microstructure thus can be regarded as decisive to understand for reducing uncertainty of future POC fluxes.

#### 20 1 Introduction

In the euphotic zone of the oceans, primary producers and subsequent grazers fuel the particulate organic matter (POM) pool that can sink in form of marine particles. Their settling velocity and remineralization eventually determines, how much

<sup>&</sup>lt;sup>1</sup>Max Planck Institute for Meteorology, Hamburg, Germany

<sup>&</sup>lt;sup>2</sup>present address: Geophysical Institute and Bjerknes Centre for Climate Research, University of Bergen, Bergen, Norway

<sup>&</sup>lt;sup>3</sup>Max Planck Institute for Marine Microbiology, Bremen, Germany

<sup>&</sup>lt;sup>4</sup>Leibniz Institute for Baltic Sea Research, Rostock, Germany

<sup>&</sup>lt;sup>5</sup>now: University of Hamburg and Helmholtz Centre Hereon, Hamburg, Germany

35

particulate organic carbon (POC) escapes the euphotic zone as export production, estimated to be between  $4 \,\mathrm{Pg} \,\mathrm{Cyr}^{-1}$  to  $11.2 \,\mathrm{Pg} \,\mathrm{Cyr}^{-1}$  (Laws et al., 2000; Najjar et al., 2007; Henson et al., 2012), recently constrained to  $7.36 \pm 2.12 \,\mathrm{Pg} \,\mathrm{Cyr}^{-1}$  (Yamaguchi et al., 2024), to the mesopelagic zone of the oceans. The subsequent attenuation of POC fluxes in the mesopelagic zone determines the deep oceans fluxes and thereby the storage capacity due to the biological carbon pump (Wilson et al., 2022). This makes the investigation of processes potentially affecting export production and attenuation of POC fluxes crucial for understanding the evolution and the feedback of the biological carbon pump on the Earth system under climate change (Ilyina and Friedlingstein, 2016). Recently, Maerz et al. (2020) provided an advanced representation of the biological carbon pump in the HAMburg Ocean Carbon Cycle (HAMOCC) model in an ocean standalone setup under climatological atmospheric conditions. With the present work, we present and discuss the effects of this more realistic representation of the biological carbon pump on primary production and biogeochemical fluxes under projected future climate warming in the framework of the Max Planck Institutes Earth system model (MPI-ESM1.2-LR configuration; Mauritsen et al., 2019) of which HAMOCC (Six and Maier-Reimer, 1996; Ilyina et al., 2013; Paulsen et al., 2017) is the ocean biogeochemistry component.

Constraining the export production and fate of marine particles is afflicted with high uncertainties, partially even in the sign of response to projected future climate warming (Laufkötter et al., 2015; Henson et al., 2022). To understand export fluxes and their attenuation in the mesopelagic zone has thus led to joint efforts in studying the twilight zone (Martin et al., 2020; Henson et al., 2022). Among ESMs taking part in the Coupled Model Intercomparison Project (CMIP), the response of export production to climate warming is closely related to the represented ecosystem structure and organic matter routing in the euphotic zone and the linked process of particle formation and remineralization (Laufkötter et al., 2016). Temperature-dependent remineralization contributes to ecosystem shifts with ocean heat uptake and increased stratification under climate warming (Segschneider and Bendtsen, 2013; Crichton et al., 2021). In comparison to models with globally constant remineralization rates, diatoms experience higher silicate limitation and calcifiers are favoured (Segschneider and Bendtsen, 2013). Calcification leads to loss of alkalinity with negative consequences for the oceanic carbon dioxide (CO<sub>2</sub>) sink (Planchat et al., 2023). Both, opal and CaCO<sub>3</sub>, can act as ballasting material for POC (Armstrong et al., 2002; Passow, 2004) and can increase the POC transfer efficiency through the mesopelagic zone (Klaas and Archer, 2002; Balch et al., 2010; Cram et al., 2018). In addition to gravitational sinking, active vertical migration of higher organisms with their fecal pellet production are expected to contribute to vertical POC fluxes (Archibald et al., 2019). These active biological fluxes have yet to be considered in CMIP6 models (Henson et al., 2022). Zooplankton vertical migration (Bandara et al., 2021) potentially increases global POC export flux by about 14 % to 20 % (Aumont et al., 2018; Archibald et al., 2019; Pinti et al., 2023) which thus induces potential biases in present CMIP6 ESMs. Physical mechanisms, like the eddy pump (Boyd et al., 2019) and sub-mesoscale to mesoscale eddy-driven particle size-dependent export (Lévy et al., 2012; Dever et al., 2021) additionally affect POC fluxes with partially uncertain response in a future climate. Molecular sea water viscosity decreases with warming oceans in future which can lead to enhanced sinking velocities, while its overall effect on future POC fluxes is with about 3\% on export production likely minor (Taucher et al., 2014; Henson et al., 2022). Attenuation of sinking fluxes ultimately occurs through remineralization of POM whose underlying microbial processes are temperature-dependent (Dell et al., 2011). Since remineralization is energetically favourable under oxic conditions, oxygen deficit zones (ODZs) hamper remineralization and thus attenuation of POC fluxes

75

(Le Moigne et al., 2017; Weber and Bianchi, 2020; Cram et al., 2022). This makes ODZs to regions of high transfer efficiency of exported POC to greater depth (Roullier et al., 2014; Weber et al., 2016; Maerz et al., 2020; Weber and Bianchi, 2020) while being harmful to higher organisms (Heinze et al., 2021). Recently, fragmentation of larger particles has been recognized as potential mechanism that contributes to longer residence time of generated smaller particles in the water column (Briggs et al., 2020), which allows for stronger decline of POC fluxes through ongoing remineralization. Particularly in the wind-driven mixed layer of the ocean, turbulence-induced shear can fragment particles (Takeuchi et al., 2019). In addition, zooplankton can break up particles while swimming (Dilling and Alldredge, 2000) or handling of particles (Mayor et al., 2014, 2020). Aggregation processes of particles, but also their susceptibility to fragmentation, are linked to the particle compounds, the 'primary particles', and the particles internal microstructure, which is affected by the compounds size and stickiness (Meakin, 1988; Song et al., 2023) or the generating process (Jiang and Logan, 1991), for example through zooplankton egestion (Kilps et al., 1994). Eventually, particle size and density, and to certain extent shape, determine the terminal sinking velocity in non-stratified waters (Ahmerkamp et al., 2022).

The representation of the complex processes in a heterogeneous ocean (Azam and Malfatti, 2007) to great detail in ESMs is unrealistic due to both, large process uncertainties and high computational costs. Hence, only climate-relevant processes are represented in an often aggregated or simplified form. However, simplified representations tend to be stiffer in their response to climate change induced perturbations, while more comprehensive ones potentially respond more realistically and can thus better capture e. g. local variability and interact with a changing ocean biogeochemistry state in a variable climate.

The generally heterogeneous sources of particles and the transformation and loss processes that they can undergo, make it challenging to study their dynamics in situ and to represent in models. Dedicated model studies contribute to understand the potential implications of marine particles on both, ocean food webs and biogeochemistry (e.g. Jackson, 1990; Kriest and Evans, 2000; Armstrong et al., 2002; Stemmann et al., 2004b, a; Jackson and Burd, 2015; Jokulsdottir and Archer, 2016; Gloege et al., 2017). Uncertain parameters and high computational costs typically hinder to include processes to greater detail in ESMs (Dinauer et al., 2022). In ESMs, simplifications for gravitational particle sinking range from constant sinking velocity (Ilyina et al., 2013), power law assumptions of POC fluxes (Martin curve; Martin et al., 1987; Kriest and Oschlies, 2008; Mauritsen et al., 2019), to more advanced sinking schemes, where POM quality (Aumont et al., 2017), ballasting (Heinemann et al., 2019; Karakuş et al., 2025; Armstrong et al., 2002; Stock et al., 2020, the latter two including mineral protection of POC), or particle size distributions are considered (Kriest and Evans, 1999, 2000; Gehlen et al., 2006; Schwinger et al., 2016). However, the heterogeneity and structural complexity of marine particles and related microbial dynamics become more and more acknowledged to be important for determining POC fluxes (Omand et al., 2020; Nguyen et al., 2022). With the recently developed M<sup>4</sup>AGO (Microstructure, Multiscale, Mechanistic, Marine Aggregates in the Global Ocean) sinking scheme, Maerz et al. (2020) explicitly represent the heterogeneous composition, microstructure and size spectrum of marine particles in combination with a temperature- and oxygen-dependent remineralization rate. M<sup>4</sup>AGO thus explicitly disentangles compositional and structural effects on the sinking of marine particles. This comprehensive approach makes it a valuable representation to consider for climate change projections in order to provide insights into potential responses of biogeochemical fluxes.

With the present study, we i) present and discuss effects of the more comprehensive particle representation via the M<sup>4</sup>AGO-extended MPI-ESM1.2-LR on net primary production and CO<sub>2</sub> fluxes compared to the standard CMIP6 (Coupled Model Intercomparison Project phase 6) model version of MPI-ESM1.2-LR (Paulsen et al., 2017; Mauritsen et al., 2019) and ii) investigate implications for future particle properties and ocean biogeochemical fluxes.

#### 2 Methods

#### 2.1 Brief model description

For the present study, we extended HAMOCC (Six and Maier-Reimer, 1996; Ilyina et al., 2013; Paulsen et al., 2017; Mauritsen et al., 2019) as part of the MPI-ESM1.2-LR (Mauritsen et al., 2019) with the M<sup>4</sup>AGO sinking scheme (Maerz et al., 2020). The low resolution (LR) setup of MPI-ESM1.2 refers to a nominal Max-Planck-Institute Ocean and sea ice Model (MPIOM; Marsland et al., 2003; Jungclaus et al., 2013) resolution of 1.5 ° and 40 uneven vertical layers with finer resolution in the near-surface. The ocean component is coupled to the atmospheric component ECHAM 6.3 and experiences riverine fresh water discharge from land via the hydrological model (Hagemann and Dümenil, 1997; Mauritsen et al., 2019).

For a detailed description of M<sup>4</sup>AGO, please refer to Maerz et al. (2020) who evaluated the novel sinking scheme and its effects on the global biogeochemistry in a MPIOM standalone setup under climatological atmospheric conditions (Röske, 2005). In brief, M<sup>4</sup>AGO explicitly represents spatio-temporally variable, heterogeneously composed marine particles that feature a microstructure and power law size spectrum with slope b. The particles are composed of primary particles which exhibit an attributed size, density and stickiness. Primary particle numbers are calculated from the local concentration of particle-forming HAMOCC tracers, namely detritus, opal, CaCO<sub>3</sub> and dust. The mean particle stickiness heuristically defines the microstructure of particles, the fractal dimension  $d_f$ . The number of primary particles and their attributed properties in combination with the fractal dimension enables to calculate mean primary-particle density and size,  $\langle \rho_p \rangle$  and  $\langle d_p \rangle$ , respectively. Fractal dimension, together with the mean primary particle size, defines a variable particle porosity throughout the size spectrum that extends from  $\langle d_p \rangle$  to the maximum diameter  $d_{\text{max}}$  which is defined through a critical particle Reynolds number. In summary, M<sup>4</sup>AGO distinguishes between structure and composition based on sub-particle scale properties which leads to variable particle excess densities and sinking velocities. In the MPI-ESM1.2-LR M<sup>4</sup>AGO setup, the particle-forming tracers eventually sink with the locally derived mass concentration-weighted mean sinking velocity.

In the MPI-ESM setup with  $M^4AGO$ , opal dissolution and aerobic, oxygen concentration-dependent detritus remineralization are temperature-dependent, following a  $Q_{10}$  approach with a  $Q_{10}$  factor of 2.6 and 2.1, respectively. In the CMIP6 standard version of HAMOCC, detrital POM features a sinking velocity that is constant in the euphotic zone,  $z_{eu}$  ( $z_{eu}$  defined as being 100 m in HAMOCC) and linearly increases with depth below, following a globally uniform Martin curve (Martin et al., 1987) representation according to Kriest and Oschlies (2008). In the standard setup, the aerobic remineralization rate is only oxygen concentration-dependent and opal dissolution depends linearly on temperature. For further details of the CMIP6 MPI-ESM1.2-LR standard model, refer to Mauritsen et al. (2019). In both model setups, CaCO<sub>3</sub> can dissolve below the dynamically evolving lysocline, while dust is transported inert to chemical reactions after aeolian deposition. In contrast to  $M^4$ AGO, the

CMIP6 version features constant, independent sinking velocity for dust, opal and CaCO<sub>3</sub> and thus no variability in their sinking speeds.

#### 2.2 Experimental model setup, spinup & analysis

For the setup of the CMIP6 standard version of HAMOCC, refer to Mauritsen et al. (2019). CMIP6 MPI-ESM1.2-LR output is publicly available through the ESGF portal (Cinquini et al., 2014).

As initialization for the MPI-ESM1.2-LR M<sup>4</sup>AGO setup, we adopted the well spun-up M<sup>4</sup>AGO-MPIOM HAMOCC restart output (in total, about 1700 years of simulation time with climatological forcing, Maerz et al., 2020) and adjusted the inventories for phosphate, nitrate, dissolved inorganic carbon (DIC) and total alkalinity,  $A_T$ , according to the CMIP6 MPI-ESM1.2-LR standard model. We kept the silicate inventory untouched, since i) the CMIP6 run exhibits a too high silicate inventory, and ii) in contrast to the CMIP6 standard version, opal production affects POM sinking velocity in M<sup>4</sup>AGO. We combined the adjusted HAMOCC restart outputs with the physical spun-up state of the MPI-ESM1.2-LR standard pre-industrial control (piCtrl) run. To allow for an adjustment of the biogeochemical tracer distribution to the MPI-ESM1.2-LR ocean circulation that is different from the climatological forced MPIOM circulation, we spun up the model for about 700 years under pre-industrial climatological atmospheric CO<sub>2</sub> concentration (284.3 ppm), nitrogen deposition (Hegglin et al., 2016) and climatological dust deposition (Mahowald et al., 2005). During the spin-up time, we adjusted the weathering rates to account for loss of POC, CaCO<sub>3</sub> and silicate due to sediment burial. The final weathering rates were then kept constant (here provided for non-leap years with 365 days:  $107.2 \,\mathrm{Gmol}\,\mathrm{P}\,\mathrm{vr}^{-1}$  dissolved phosphorus,  $4.54 \,\mathrm{Tmol}\,\mathrm{Si}\,\mathrm{vr}^{-1}$  dissolved silicon, and  $30.3 \,\mathrm{Tmol}\,\mathrm{C}\,\mathrm{vr}^{-1}$  as DIC and the corresponding amount to surface alkalinity as DIC:2  $A_T$ ) during the subsequent piCtrl run under the same pre-industrial CO<sub>2</sub> forcing, in which negligible drifts in surface variables were visible (year mean  $CO_2$  oceanic uptake:  $0.029\,\mathrm{Tg}~\mathrm{Cyr}^{-1}$ , trend for uptake:  $0.009 \,\mathrm{Tg} \,\mathrm{Cyr}^{-1} \,\mathrm{century}^{-1}$ ). No further parameter adjustments were carried out. Starting from the piCtrl state and keeping the weathering rates constant, we ran the MPI-ESM1.2-LR M<sup>4</sup>AGO setup with the CMIP6 historical forcings for the years 1850 to 2014, followed by the high-emission shared socio-economic pathway (SSP) scenario SSP5-8.5 (Riahi et al., 2017) for the time period 2015 to 2099. For climatological means, we refer to the 30-year time period 1985 to 2014 as historical and 2070 to 2099 as future period, if not otherwise stated. Due to the spin-up procedure, the physical ocean dynamics is not bit identical and year-to-year differences due to internal variability can be expected. However, the physical climatological mean state for both the historical and future period is not or only minor affected by the different spin-ups of the CMIP6 and M<sup>4</sup>AGO model version, so that a comparison between the two versions provides meaningful insights into the structural differences of the biogeochemistry model. Hence, where applicable, we compare the results with the M<sup>4</sup>AGO scheme with a run of MPI-ESM1.2-LR carried out for the Coupled Model Intercomparison Project phase 6.

#### 3 Results & Discussion

The results and discussion section is divided into two main parts. In Sec. 3.1 to 3.3, we provide and discuss the large scale pattern and compare the MPI-ESM1.2-LR CMIP6 (hereafter: CMIP6) to the MPI-ESM1.2-LR M<sup>4</sup>AGO (hereafter: M<sup>4</sup>AGO)





model version. In Sec. 3.4 to 3.5, we investigate and discuss the consequences of changing export fluxes in M<sup>4</sup>AGO on marine particle dynamics, flux attenuation and consequently on the biological carbon pump. Each subsection is subdivided into a results-oriented and discussion part. This section is completed with a general discussion (Sec. 3.6).

# 160 3.1 Primary and export production

Large scale circulation pattern and vertical mixing determine the availability of nutrients in the euphotic zone of the oceans and thus co-determine, together with light availability, primary production. For a detailed analysis of the ocean dynamics in MPI-ESM refer to Jungclaus et al. (2013). Ongoing ocean warming tends to increase stratification and to decrease vertical mixing, thus mixed layer depths, which impinges on nutrient availability in the euphotic zone and thus net primary production. By contrast, ongoing summer sea ice loss, particularly in the Arctic Ocean, increases light availability for primary producers and the susceptibility to wind-driven mixing which lowers stratification.

For the historical period, the global depth-integrated net primary production (NPP) in the  $M^4AGO$  run is with  $55.0~\rm Gt~Cyr^{-1}$  about  $7.9~\rm Gt~Cyr^{-1}$  higher than the CMIP 6 run ( $47.1~\rm Gt~Cyr^{-1}$ ). Particularly in the low latitudes,  $M^4AGO$  exhibits higher NPP than the CMIP 6 run (Fig. 1). Until the end of the century, the simulations show a global decline in NPP of  $\approx 9.7~\%$  and  $\approx 8~\%$   $M^4AGO$  for CMIP6, respectively. The zonal response, however, differs and  $M^4AGO$  shows with -15.8~% a 2.4~% larger decline of NPP in the low latitudes ( $30^\circ S$  to  $30^\circ S$ ) compared to the CMIP 6 run (-13.4~%). In the southern mid-latitudes ( $66^\circ S$  to  $30^\circ S$ )  $M^4AGO$  shows a lower decline in NPP, while a stronger decline of NPP in the northern mid-latitudes ( $30^\circ N$  to  $66^\circ N$ ; -8.1~% versus -6.3~% in the CMIP 6 run). In the northern latitudes ( $>66^\circ N$ ), NPP is generally low due to the ice cover at present day, seasonality and generally smaller areal extend compared to the global ocean, but increases with continued warming by 20.3~% in the  $M^4AGO$  run and with 26.9~% even more in the CMIP 6 run.

The loss of biologically bound CO<sub>2</sub> through gravitational POC fluxes,  $F_{POC}(z)$ , at the euphotic depth  $z=z_{eu}$  is considered as export production. We here use the commonly applied standard of  $z_{eu}=100\,\mathrm{m}$  as export depth. In the historical period, global  $F_{POC}(z_{eu})$  differs between the two models and the CMIP6 version features with  $\approx 5.89\,\mathrm{Gt}$   $\mathrm{Cyr}^{-1}$  a higher export production than  $\mathrm{M}^4\mathrm{AGO}$  with  $\approx 5.36\,\mathrm{Gt}$   $\mathrm{Cyr}^{-1}$ . Under the SSP5-8.5 scenario, global export productions decrease by  $\approx 11.8\,\%$  and  $\approx 12.6\,\%$  to  $\approx 5.2\,\mathrm{Gt}$   $\mathrm{Cyr}^{-1}$  and  $\approx 4.68\,\mathrm{Gt}$   $\mathrm{Cyr}^{-1}$  for CMIP6 and  $\mathrm{M}^4\mathrm{AGO}$ , respectively. The general pattern of the fluxes is qualitatively consistent between the two model versions and follow the primary production (cmp. Fig. 2 with Fig. 1; see also Appendix C for POC and biogenic mineral fluxes in  $\mathrm{M}^4\mathrm{AGO}$ ). Export fluxes, including also the mineral components opal,  $\mathrm{CaCO}_3$  (and dust), and their ratios determine the particle properties in  $\mathrm{M}^4\mathrm{AGO}$ . In Sec. 3.4, we thus discuss the impact of future export flux changes on particle properties in  $\mathrm{M}^4\mathrm{AGO}$ .

POC export fluxes are often set in relation to depth-integrated NPP. The so-called export efficiency (p ratio)

$$p \operatorname{ratio}(z) = \frac{F_{POC}(z)}{NPP} \tag{1}$$

provides the fraction of NPP-bound CO<sub>2</sub> and associated nutrients lost from the euphotic zone.

Noticeable, the CMIP6 model version features a strong latitudinal pattern of the p ratio, with values up to about 0.3 in the subtropical gyres regions (maximum  $\approx$ 0.34) and lower values in extra-tropical regions (Fig. 3) ranging globally around

**Figure 1.** Depth-integrated net primary production (NPP). Zonally integrated and maps for the historical period and anomalies in the future period. a) Zonally integrated NPP in the historical and the future period for the CMIP 6 and M<sup>4</sup>AGO run. b) Absolute changes of the zonally integrated NPP for the CMIP 6 and M<sup>4</sup>AGO run between the future and historical period. c) Percentage changes for the zonally integrated NPP between the future and historical period. The regional absolute and percentage values refer to the latitudinal regions indicated by the grey lines. d) and e) Vertically integrated NPP in the historical period for the CMIP6 and M<sup>4</sup>AGO model version. f) and g) Changes of climatological mean NPP in future projections relative to the historical period.



**Figure 2.** a), c) Climatological POC export fluxes for the historical period (1985-2014) and b), d) their difference between the future projection (2070-2099) and the historical period for CMIP6 and M<sup>4</sup>AGO.

0.13±0.04. By contrast, M<sup>4</sup>AGO shows a less pronounced and more balanced latitudinal picture and the p ratio ranges typically around 0.10±0.01. Particularly in the tropical and subtropical regions, the p ratio response to future climate warming differs and the model versions even show opposite signs in their anomalies (Fig. 3). The CMIP6 model version shows positive future p ratio anomalies in the eastern equatorial and subtropical regions, while in the M<sup>4</sup>AGO version, the p ratio declines in that region. In the high latitudes, the two model versions agree in the sign and p ratios tend to increase in both model versions.
 Overall, the response in M<sup>4</sup>AGO seem to be more buffered to future changes and p ratios experience lower changes than the CMIP6 model.

Both, the absolute differences in NPP, export fluxes and p ratio, but also the response to projected future warming of the two model versions are related to their process representation. In contrast to CMIP6,  $M^4$ AGO features a temperature-dependent remineralization and a dynamically evolving, primarily latitudinally variable sinking velocity of marine particles. Globally, the two model versions show similar NPP response to the future projection. At the latitudinal and regional scale, however, their different process representations imprint on the response to future warming. Generally, temperature is expected to be a major driver of ecosystem dynamics through its effect on enzymatic kinetics and thus growth, respiration and remineralization processes (Dell et al., 2011).  $M^4$ AGO represents the temperature-dependent POM remineralization with subsequent consequences for particle properties and thus sinking velocity (see Sec. 3.4). The higher remineralization particularly in the tropical and subtropical regions featured by  $M^4$ AGO cause the higher NPP and the lower p ratio compared to CMIP6 and being globally close to recent estimates of  $53 \pm 7$  Gt Cyr $^{-1}$  (Johnson and Bif, 2021). Rising water temperatures with ongoing future warming tend




Figure 3. a), c) Climatological export efficiency (p ratio) for the historical period (1985-2014) and b), d) its difference between the future projection (2070-2099) and the historical period for CMIP6 and  $M^4$ AGO.

to strengthen stratification and the concurrent weaker mixing recovers less of the exported nutrients. As a consequence, higher p ratios can be expected to result in higher loss of nutrients and thus lower NPP with increasing stratification. This is particularly the case in the tropical and subtropical regions for the CMIP6 model. By contrast, even though M<sup>4</sup>AGO shows increasing sinking velocities (see Sec. 3.4), the increasing remineralization in M<sup>4</sup>AGO buffers the effect of increasing stratification. This is visible in the p ratios decline in future in these regions. Despite this, we see a stronger decline in nitrate concentration in the Panama basin in the course of the future projection, likely caused by increased oxygen deficit zones (ODZs) in M<sup>4</sup>AGO (see also next section 3.2) which can cause the stronger NPP decline. As a result, this leads to slightly stronger relative reduction of NPP (15.8%) in tropical and subtropical regions compared to CMIP6 (13.4%; Fig. 1), while the buffering through lower pratios and their changes is similar to findings of Segschneider and Bendtsen (2013) for temperature-dependent remineralization. While the two model versions significantly differ in their latitudinal p ratio pattern, neither of them features the high p ratios in high latitudes as suggested by observation-based estimates while large uncertainties still persist (see DeVries and Weber, 2017). The lower p ratios in the high latitudes potentially affect  $CO_2$  fluxes and sequestration in both, the historical and future period. In the Arctic Ocean, the prominent future increase in both, NPP and p ratios in both simulations can be explained by complete sea ice loss in summer month and concurrent higher light availability for primary producers. Nonetheless, difference of relative latitudinal response is most noticeable for the Arctic Ocean. While both models agree in their sign, M<sup>4</sup>AGO only suggests a 20.3 % increase compared to 26.9 % in CMIP6 while the p ratios are similar or even higher in CMIP6. In addition to changes in p ratios, the deep sequestration of nutrients below the winter mixed layer depth (MLD) contributes to





the response. M<sup>4</sup>AGO exhibits a higher POC transfer efficiency in high latitudes (Maerz et al., 2020, see also next section) and hence associated nutrients are thus sequestered deeper. In comparison to the Southern Ocean, vertical mixing is weaker in the Arctic Ocean and sequestered nutrients are less effectively mixed back or transported into the euphotic zone. Despite future weakening of Arctic Ocean stratification during summer months, this can contribute to a weaker increase of NPP in M<sup>4</sup>AGO (see also Sec. 3.5).

In summary, the global response of both model versions to future climate warming are comparable to other ESMs that show changes of NPP of about -15% to 30% under high emission Representative Concentration Pathway scenario RCP8.5 (Laufkötter et al., 2015) and of export fluxes by about -41% to 1.8% for the SSP5-8.5 scenario (Henson et al., 2022).

#### 3.2 Future transfer efficiency changes associated to oxygen deficit zones and Arctic warming

Export production can be transferred to greater depth, where longer sequestration of nutrients, POC and ultimately  $CO_2$  can take place. Deep sequestration potential is often expressed as mesopelagic transfer efficiency that provides an aggregated measure on which exported fraction reaches greater depth, typically  $z = 1000 \,\mathrm{m}$ ,

$$T_{\text{eff}}(z) = \frac{F_{\text{POC},z}}{F_{\text{POC},z_{\text{corr}}}} \tag{2}$$

here calculated for the model grid-defined 960 m depth instead of the 1000 m horizon (Fig. 4).

In the historical period, the transfer efficiency in M<sup>4</sup>AGO exhibits a latitudinal pattern with higher values in the high latitudes compared to subtropical gyre regions (Fig. 4 a). High transfer efficiencies are also found in upwelling areas of low and mid latitude regions, where oxygen deficit zones prevail in the mesopelagic. Under oxygen limitation, POC remineralization rates are substantially lower than in oxygenated waters which increases transfer efficiency in ODZs.

In the future period (2070-2099), the largest effects on transfer efficiency are associated to regional changes of vertical ODZs extension (cmp. Fig. 4b with Fig. 5). Ocean warming generally decreases oxygen solubility and thus leads to reduced oxygen concentrations in the future (Schmidtko et al., 2017) which contributes to ODZs developments. The ODZs further respond to changes in NPP and export fluxes. As a consequence, the future decline of NPP leads to decreasing ODZs in upwelling regions. This is well visible along the south and north American Pacific coast, in the African Atlantic upwelling areas and the northern Indian Ocean where decreasing ODZs reduce the transfer efficiency. Note the small changes in the central Panama Basin can be attributed to persistent extended ODZs thoughout both periods. West of the Panama basin, however, vertical ODZ fraction expands in the mesopelagic (Fig. 5) which increases transfer efficiency. This behavior is likely associated to future shallower MLD and increasing stratification in these areas which decrease oxygen ventilation. By contrast, the westward extension of ODZs decreases in future. In the high latitudes, the Arctic Ocean shows a decreasing transfer efficiency in future. Here, the decline of transfer efficiency is associated to the increase in NPP (Fig. 1) and the associated higher POM export fluxes (see also Appendix C). The additional POM increases the buoyancy of marine particles on seasonal average and thus decreases settling velocity. In combination with temperature-enhanced remineralization, this decreases future transfer efficiency in the Arctic Ocean in M<sup>4</sup>AGO.

**Figure 4.** M<sup>4</sup>AGO transfer efficiency of POC. a) Mesopelagic transfer efficiency in the historical period. b) Change of the transfer efficiency in future with respect to the historical period. c) Regional mean and standard deviation comparison to present day calculation of Weber et al. (2016). CMIP6 run for comparison. d) Interannual variability of the transfer efficiency during the historical period expressed as annual standard deviation and e) changes of the interannual variability of the transfer efficiency in the future period with respect to the historical time period. Note that the pattern of the CMIP6 model version is comparable to the ocean-only setup in Maerz et al. (2020).

Figure 5. Change of vertical extent of the mesopelagic ODZ water column fraction in the  $M^4AGO$  simulation. Positive values imply greater and negative values smaller vertical extent of ODZs in the future than in the historical period. Units are in vertical meter extend of waters with  $O_2 






Maerz et al. (2020) showed that transfer efficiency varies on seasonal time scales, particularly in high latitudes. We find that it also varies interannually due to changes in i) export ratios, ii) positions of circulation pattern and associated frontal regions, and iii) internal variability of e.g. temperature and mixing with effects on remineralization and ventilation of ODZs. The variability of transfer efficiency potentially affects the biologically induced CO<sub>2</sub> drawdown and hence variability of CO<sub>2</sub> fluxes. The largest interannual variability of transfer efficiency is associated to the most dynamical ocean regions, i.e. the Southern Ocean and particularly in the Weddell and Ross Sea (Fig. 4c). Noticeably, ODZs also feature an increased interannual variability of transfer efficiency compared to well-ventilated open ocean regions. In future, the interannual variability tends to decrease in the Southern Ocean and in some of the upwelling regions, where ODZs expansion declines in the mesopelagic.

Generally, the M<sup>4</sup>AGO scheme reproduces well the transfer efficiency pattern found by Weber et al. (2016), based on diagnosed phosphate fluxes and inverse modelling, and by others (DeVries and Weber, 2017; Cram et al., 2018; Dinauer et al., 2022; Sulpis et al., 2023). Studying and gaining insights into annual global transfer efficiency patterns aids in bridging between short and long time scales. Observational time scales for fluxes are typically of an order of weeks or month to seasons, while their localization and high dependence on environmental conditions make it difficult to directly compare them with ESMs and thus to project them on longer time scales. On the long term (O(1000 yr)), these fluxes determine nutrient and DIC distributions in combination with the underlying circulation pattern. Since POC and associated nutrient fluxes and their attenuation determine annual transfer efficiency, gaining annual mean transfer efficiency from observational studies would be a valuable link between short time scales and long term nutrient distribution.

Simulated future changes in transfer efficiency as such cannot directly be related to changes of biologically induced  $CO_2$  fluxes, since transfer efficiency only expresses a ratio and not absolute amount of fluxes and thus of deep  $CO_2$  drawdown. Nevertheless, strong future changes of transfer efficiency are associated to changing NPP and export ratios (as in the Arctic). Further, any future change of extensions and thickness of ODZs affects the transfer efficiency. We therefore briefly provide insights into the ODZs evolution over the full historical and future simulation. As summarized in Maerz et al. (2020),  $M^4$ AGO features positive feedbacks on ODZs development. Namely, the lower remineralization and thus higher detritus accumulation in ODZs lead to decreased sinking velocities, since the POC to ballast ratio increases. This contributes to the overestimation of ODZs, which generally can be attributed to sluggish circulation and too little mixing particularly in equatorial regions (Aumont et al., 1999; Dietze and Loeptien, 2013; Kuntz and Schrag, 2020; Duteil et al., 2021). We therefore follow the strategy of Bindoff et al. (2019) and compare the relative changes of ODZ volume over time to enable comparison of the response of ODZs to changing environmental conditions under ongoing climate warming. We compare the changes of global ODZs size with  $O_2$  below  $20 \, \mu \text{mol} \, L^{-1}$  ('suboxic'), where significant reduction of remineralization rates are simulated in the model with the formerly discussed consequences on transfer efficiency, and a threshold of  $80 \, \mu \text{mol} \, L^{-1}$  ('hypoxic'). The latter encompasses the former and is relevant for higher organisms that are already affected by oxygen levels lower than  $80 \, \mu \text{mol} \, L^{-1}$  (Heinze et al., 2021, and references therein).

The relative ODZ core volume evolution for  $O_2 < 20\,\mu\mathrm{mol}\,\mathrm{L}^{-1}$  show a decline in the future scenario, particularly in the mesopelagic ocean (Fig. 6a). The global as well as the deep ocean ODZs respond similar in both simulations, showing a decline of about 2%, while in the mesopelagic, the M<sup>4</sup>AGO simulation shows a stronger decline and features a relatively




**Figure 6.** Relative change of ODZ volumes in CMIP6 and M4AGO. For both simulations and each examined depth horizon, the drift of the piCtrl simulation (calculated relative to the mean of time period 1850-1900) was subtracted and the result normalized to the mean volume in the period 1850-1900. Note that the pre-industrial control simulation in M<sup>4</sup>AGO showed an increase of global ODZ size due to increasing deep ocean ODZs evolution, while CMIP6 showed a decline. The mesopelagic refers to 100-1000 m and deep ocean region to ocean depths larger 1000 m. Note the different y-axis.

smaller core ODZ by the end of the future scenario (about 16% loss compared to 13% in CMIP6). In absolute terms, however, M<sup>4</sup>AGOs ODZ is still larger than in the CMIP6 simulation. The slightly stronger decline in M<sup>4</sup>AGO could be associated to the different p ratio. Lower p ratio, a stronger declining NPP in subtropical and tropical regions can lead to lower sustaining of ODZs volumes in the mesopelagic. The response of the relative global hypoxic volume for the mesopelagic is less pronounced and exhibit only a slight (CMIP6) or even no (M<sup>4</sup>AGO) decline (Fig. 6b). In contrast to the evolution of ODZ with oxygen concentration lower than  $20\,\mu\text{mol}\,\text{L}^{-1}$ , the relative hypoxic volume increases by about 1.5% and 2% to 4% globally and in the deep ocean, respectively. In summary, the ODZs core with  $O_2 < 20\,\mu\text{mol}\,\text{L}^{-1}$  tend to shrink, while the hypoxic volumes show less of a decline and even increase in the deep ocean. The global trends with contracting suboxic ODZs and expanding hypoxic ODZs are in line with other ESMs of the CMIP6 cohort which show similar trends for the Pacific Ocean (Busecke et al., 2022). Ensemble simulations with M<sup>4</sup>AGO could shed light on the significance of the different mesopelagic ODZs core trend compared to the CMIP6 simulation and provide insights, in how far not only the p ratio, but also the internal dynamics and feedbacks between sinking and remineralization drive the ODZs evolution.

## 3.3 Historical and future evolution of CO<sub>2</sub> fluxes

To date, oceanic yearly mean CO<sub>2</sub> fluxes are afflicted with high uncertainties and generally, ESMs are challenged to represent these fluxes well (Gruber et al., 2023). On average, CO<sub>2</sub> fluxes are largely driven by the global circulation field and related physical transport and mixing of carbon, mainly in the form of DIC, across the mixed layer interface (Levy et al., 2013).








In most ocean regions, the gravitational POC flux across the mixed layer interface is small compared to absolute values of these physics-driven DIC fluxes (Levy et al., 2013). However, the physical solubility pump accounts for only about 17% to 40% to the present day observed global DIC gradient, while the carbonate and soft-tissue pump account for about 21% and 62%, respectively (Bacastow and Maier-Reimer, 1990; DeVries, 2022). The changed transfer efficiency pattern between the CMIP6 and the  $M^4$ AGO model have an effect on the DIC gradients and seasonal dynamics of nutrient availability for primary production, both imprinting on  $CO_2$  fluxes. Thus, we here compare the two simulations to the mean of the harmonized  $CO_2$  flux products by Fay et al. (2021) based on three wind (Atlas et al., 2011; Hersbach et al., 2020; Kobayashi et al., 2015) and six different surface ocean  $pCO_2$  products (Chau et al., 2020; Denvil-Sommer et al., 2019; Gregor et al., 2019; Rödenbeck et al., 2013; Iida et al., 2020; Landschützer et al., 2014, 2020; Zeng et al., 2014) for the time period January 1990 to end of 2019. Internal variability in the simulations and the mean of the observational products prevents from a direct comparison with the product. We thus calculate the 30 year trend based on monthly mean values to reduce the effect of seasonal to decadal scale internal variability on the analysis. We perform a trend comparison and significance testing. For testing i) the significance of trends and ii) the significance of trend differences, we follow Santer et al. (2000) and apply their methods in combination with the adjusted standard error and the adjusted degrees of freedom.

Both simulations show a negative trend of CO<sub>2</sub> fluxes in southern high latitudes (Fig. 7a,b). In mid and low latitudes, the pictures is less clear. The CMIP6 simulation indicates a slightly increasing outgassing over the 30 year period and decrease in the eastern equatorial region, while in M<sup>4</sup>AGO, the CO<sub>2</sub> fluxes remain neutral or even indicate a negative trend, yet outgassing (see below). However, only few areas show the significance of those trends, for example the eastern equatorial Pacific in the CMIP6 simulation and boundaries of the high productive equatorial Pacific region in M<sup>4</sup>AGO. The trends of the mean CO<sub>2</sub> product show a clear latitudinal pattern with negative trends in the North Pacific and North Atlantic, negative trends in the Southern Ocean polar frontal region and positive trends in the mid and low latitudes (Fig. 7d). A larger area fraction of those areas show significance in these trends than the simulations. Comparing the simulation trend differences to the standard deviation of trends across the different CO<sub>2</sub> products, most of the differences are within the standard deviation (i.e. within the range {-1,1} of the ratio) indicating the simulations rather being compliant with observed trends, while some regions like the North Atlantic and Kuroshio region show significant trend differences between the simulations (Fig. 7c). When comparing the trend differences between the simulations and the observational product, particularly the coastal regions shine up as being significantly different in their trends as well as parts of the Arctic Ocean, and areas of the mid and North Atlantic and some areas in the eastern Pacific (Fig. 7e,f). The coastal region trend differences are likely caused by the coarse resolution of the models, as coastal and shelf regions are spatially underrepresented, but also in terms of process representation. Overall and with slightly less spatial extend of trends with significant difference, M<sup>4</sup>AGO seems to slightly improve the trend representation. Nevertheless, for most areas, trends in observations are not discernible from trends in the simulations. In summary, the trend comparison highlights the fact that yet the model simulations still show significant differences in CO<sub>2</sub> fluxes and their trends for the time period 1990 to 2019 and representing CO<sub>2</sub> fluxes remains challenging.

Despite the dominating factor of circulation on CO<sub>2</sub> fluxes, the changed pattern of transfer efficiency in M<sup>4</sup>AGO compared to CMIP6 can affect the regional CO<sub>2</sub> fluxes and their evolution in future. We thus investigate the seasonal changes of zonally





**Figure 7.** Trends in CO<sub>2</sub> fluxes for a) the CMIP6 and b) the M<sup>4</sup>AGO simulation. Yellow hatching indicates significant trends or trend difference. c) Comparing the trend difference between M<sup>4</sup>AGO and CMIP6 to the standard deviation across of trends across the difference CO<sub>2</sub> products. Hatching indicates significant trend differences between M<sup>4</sup>AGO and CMIP6. d) Trend of the mean of CO<sub>2</sub> products. e) and f) trend difference between CMIP6 or M<sup>4</sup>AGO and mean observational product trend.

integrated  $CO_2$  fluxes from 1850 through 2100 with a focus on seasonal variability and changed cumulative uptake. To reduce the imprint of sub-decadal internal variability of the circulation on our results, we compare the 10-years moving standard deviation based on monthly mean values, defining the seasonal amplitude, between the two simulations and analyze the time-cumulative of zonally integrated  $CO_2$  fluxes (Fig. 8).

The seasonal variability of CO<sub>2</sub> fluxes is particularly increasing in the high latitudes (Fig. 8 a-d) due to changes in the marine carbon chemistry (increasing Revelle factor; Landschützer et al., 2018). Throughout the entire simulated historical and future time period, the seasonal amplitude in the M<sup>4</sup>AGO simulation is increased in the northern latitudes, particularly around the 50 °N latitudinal band, associated with the North Atlantic and parts of the North Pacific. By contrast, most of the latitudes south of 50 °S M<sup>4</sup>AGO shows smaller seasonal amplitudes until mid of the last century. Parts of the Southern Ocean (SO; around 60 °S) features higher and the northern SO smaller seasonal amplitudes by the end of this century. North of the SO boundary, the seasonal amplitude in CO<sub>2</sub> fluxes is increased in M<sup>4</sup>AGO and increases further in future. The SO thus undergoes a more rapid change in seasonal amplitudes of CO<sub>2</sub> fluxes than in the CMIP6 model version. By contrast, in the equatorial regions, the seasonal amplitude was enhanced in the historical period while tends to decrease compared to the CMIP6 simulation in future. When comparing the time-cumulative zonally integrated CO<sub>2</sub> fluxes, the M<sup>4</sup>AGO simulation features an increased downward flux component in the SO and northern latitude region, while featuring a stronger upward component in equatorial regions. This makes the 50 °S band becoming a net sink for CO<sub>2</sub> in the M<sup>4</sup>AGO simulation earlier and the equatorial region later than the CMIP6 model version (Fig. 8 a,b), which is in line with a deeper sequestration of biologically bound CO<sub>2</sub> in the high latitudes and shallower in the equatorial regions due to the changed transfer efficiency pattern. By end of this century, the M<sup>4</sup>AGO simulation has taken up about 2.2 Gt C less anthropogenic CO<sub>2</sub> than the CMIP6 model version, which is a small, but non-

Figure 8. Time-evolution of sea-air  $CO_2$  flux anomaly (positive into air; base are the global means of the piCtrl simulations for the entire period). a) and b) Zonally integrated monthly  $CO_2$  flux anomalies. Green and magenta lines indicate the annual zero mean flux isoline for the CMIP6 and the  $M^4AGO$  simulation, respectively, when the ocean switches from net source/sink to sink/source. In b) CMIP6 green line is drawn for comparison. c) 10-year moving standard deviation of monthly zonally integrated  $CO_2$  fluxes as indicator for the seasonal amplitude for CMIP6 and d) the difference ( $M^4AGO$  - CMIP6). e) Time-cumulative of zonally integrated  $CO_2$  fluxes for CMIP6 and f) the difference between  $M^4AGO$  and CMIP6. g) latitudinal cumulative zonal sum of the difference shown in f) for the time snap shot 12/2099. The  $\approx 2221 \, \text{Mt} \, \text{C}$  indicates an about 2.2 Gt C lower oceanic uptake by  $M^4AGO$  over the full time period. Negative values at the zero latitude indicate a reduced oceanic north-south  $CO_2$  transport in  $M^4AGO$  compared to CMIP6.

negligible lower contribution to carbon sequestration by the biological carbon pump through the changed transfer efficiency pattern.  $2.2\,\mathrm{Gt}$  C correspond to about  $0.5\,\%$  of the total oceanic  $\mathrm{CO}_2$  uptake until the end of the century or equivalently to about one year of present day oceanic  $\mathrm{CO}_2$  uptake. However, the simulation still falls into the range of the MPI-ESM grand ensemble (MPI-GA; Maher et al., 2019) that resolves internal variability to great extend. The more pronounced  $\mathrm{CO}_2$  uptake



particularly in the SO in M<sup>4</sup>AGO is likely offset by the stronger decline in net primary production in low latitudes and thus lower CO<sub>2</sub> uptake. Changes in CO<sub>2</sub> uptake over the coming decades to century are dominated by physical solubility and ocean circulation changes (DeVries, 2022; Visser, 2025). For example, this has been shown for the North Atlantic where the uptake is strongly determined by the Atlantic Meridional Overturning Circulation strength (Goris et al., 2023). In turn, difference in transfer efficiency pattern on high latitude CO<sub>2</sub> uptake has thus far not been rigorously analyzed. However, a generally small effect of the transfer efficiency pattern on transient CO<sub>2</sub> uptake is expected since the biological carbon pump only maintains DIC gradients and strong changes in ocean biogeochemistry would be required to change these gradients (see also comment of Broecker, 1991). Hence, ensemble simulations with M<sup>4</sup>AGO in comparison to the MPI-GA could provide a more detailed answer, to what extend a changed transfer efficiency pattern affects the transient response of the biological contribution to CO<sub>2</sub> uptake under climate change.

#### 375 3.4 Future particle property changes & research implications

Future changing export flux ratios affect particle composition, thus potentially particle properties and eventually vertical transport. Thus far, studies focused on individual aspects of particle properties and their environment to investigate the potential future individual effects on vertical fluxes. With M<sup>4</sup>AGO, we have the opportunity to investigate potential consequences of composition changes on particle sinking and flux attenuation in a scheme where all these individual aspects are mechanistically linked together. Considering the underlying mechanistics and process understanding of M<sup>4</sup>AGO as reasonable, we in the following aim at illuminating potential future research needs from a particle modeling perspective. In this section, it is thus our objective to provide model-based diagnosed contributions of model parameters to future mean sinking velocity changes. Higher future impacts of particle properties on sinking velocity suggest higher uncertainty in these particle properties, which thus qualify as potential future research needs.

In order to identify diagnostic model parameters that affect future settling velocity most, we define the relative contribution of changes of parameters,  $X_i = \{\langle \rho_p \rangle, \langle d_p \rangle, d_f, \mu, d_{\max}, b\}$ , to changes in mean sinking velocity as follows

$$RC_{X_i}[\%] = 100 \cdot \frac{\partial_{X_i} \langle w_s \rangle \cdot \Delta X_i}{\sum_i |\partial_{X_i} \langle w_s \rangle \cdot \Delta X_i|}$$
(3)

where  $\partial_{X_i}\langle w_s\rangle$  is the partial derivative of the mass concentration-weighted mean sinking velocity with respect to diagnostic parameter  $X_i$  and  $\Delta X_i = X_{i,\mathrm{proj}} - X_{i,\mathrm{hist}}$  is the absolute change of the climatological diagnostic parameter mean between the historical,  $X_{i,\mathrm{hist}}$ , and the future projected  $X_{i,\mathrm{proj}}$ , time period. For clarification of the diagnostic model parameters, see caption Fig. 9 and to greater detail Maerz et al. (2020). By neglecting higher order terms in the derivatives, we only provide a first order estimate of parameters influence on future changes in sinking velocity (for historical  $\langle w_s \rangle$  and absolute changes in the future projection, see Appendix D, Fig. D1).

We defined the relative contribution,  $RC_{X_i}$ , in such a way that it provides the sign of response, the relative percentage change of the historical settling velocity in the future due to parameter  $X_i$ , and  $\sum_i |RC_{X_i}| = 100\%$ . Strongest RC appear from changes in mean primary particle density,  $\langle \rho_p \rangle$ , and fractal dimension of particles,  $d_f$  (Fig. 9). Sinking velocity in future





Figure 9. Relative contribution of changes in climatological mean particle properties and dynamic viscosity between the historical (1985-2014) and future (2070-2099) period to changes in sinking velocity at export depth (Eq. (3)). The mean sinking velocity determining diagnostic model parameters are:  $\langle \rho_p \rangle$ : mean density of primary particles;  $\langle d_p \rangle$ : mean diameter of primary particles;  $d_f$ : fractal dimension of the particles (microstructure);  $\mu$ : molecular dynamic viscosity of sea water;  $d_{\text{max}}$ : maximum particle diameter; b: the slope of the particle number distribution

response less to dynamic molecular viscosity,  $\mu$ , and changes in mean primary particle size,  $\langle d_{\rm p} \rangle$  and even lesser to changes in maximum particle diameter,  $d_{\rm max}$ , and the number distribution slope, b.

The changes in these diagnostic particle properties and their individual contributions are closely connected to changes in concentration of tracers that sink together and thus to export fluxes (see Appendix C and Fig. C1 therein). Mean primary particle density,  $\langle \rho_p \rangle$ , affects the sinking velocity most, particularly in the tropical, subtropical and North Pacific region, and is driven by changes in the CaCO3 to POC export ratios making ballasting CaCO3 a strong driver. In many of those regions, the microstructure changes as well and the particles become more compact, visible in higher fractal dimension and their respective positive RC. The fractal dimension also affects the mean primary particle density, so their largely co-varying pattern can be expected. In the silicate-deprived regions, the increased CaCO3 to POC ratio and thus ballasting with CaCO3 makes particles more prone to fragmentation and repacking, generating more compact particles with higher fractal dimension, and thus decreases maximum particle diameters in the future. In regions, where the opal to POC export flux ratio increases in the future period, mean primary particle size increases which contributes to increased sinking velocities. In the high latitude regions, where a transition from frequently ice-covered or at least ice-influenced to seasonally ice-free happens in future, increased NPP (see Sec. 3.1), and thus detritus production, leads to, on average, looser, less compact, larger particles. The increase of water temperature in most parts of the euphotic zone and upper mesopelagic of the oceans leads to lower dynamic sea water viscosity and thus contributes to increasing sinking velocities.

The tendencies drawn from the diagnostically evolving particle properties and their physical environment generally agree with former studies. Production of ballasting material has been suggested to decrease in future and thus its effect on export fluxes (Henson et al., 2022). In HAMOCC with M<sup>4</sup>AGO, ballasting fluxes are projected to decline in equatorial regions, and






slightly increase in subtropical regions (see Appendix C). In these regions, particles composition shift towards a more ballast-affected particle regime with denser mean primary particles and being more compact. This leads to increased sinking velocities. The increased sinking velocity cannot counterbalance the increased POC remineralization, leading in sum to reduced export fluxes of POC compared to NPP which is visible in the reduced p ratio in future (Sec. 3.1). Temperature-dependent remineralization has thus a two fold effect on particles in  $M^4AGO$ : i) an increase in primary particle density and ii) decreasing porosity, which together lead to denser, faster sinking particles. Hence, changes of the ratio between in biogenic mineral shell material and aggregated POM appear as driver for changes in settling velocity and to lesser extent, the amount of ballasting material alone. The effect on export flux is thus not purely additive and thus contrasts to approaches like in Hofmann and Schellnhuber (2009). At euphotic depth, the molecular dynamic viscosity has a relatively strong influence. Its relevance, however, decreases when considering the full water column. For the latter, it is estimated by Taucher et al. (2014) to contribute to increased sinking fluxes by about 3 % (Henson et al., 2022). Only few regions of the oceans show proportionally extremely increasing temperatures in the mesopelagic (see Sec 3.5). Among them the Arctic Ocean, where temperature increase by more than 100% in the mesopelagic, which translates to decreasing viscosity by about 8% to 10%. While the mesopelagic in the subtropics is also warming, the effect is less and extrapolating the shown relative contribution of viscosity on sinking velocity to depth would thus overestimate its contribution.

Based on a model study, Leung et al. (2021) suggested variable particle distributions in combination with temperature-dependent remineralization and temperature effects on phytoplankton size structure could lead to a negative feedback on export production due to smaller, slower sinking particles. Indeed, in subtropical regions, the modeled maximum particle size tend to decrease which is in some areas accompanied by an increasing distribution slope (more small compared to large particles), both congruent with the analysis of Leung et al. (2021). However, in M<sup>4</sup>AGO this is associated by increasing primary particle density and fractal dimension, leading to denser particles and thus faster sinking velocity. In M<sup>4</sup>AGO, the suggested negative feedback is thus at least partially offset by increasing particle density due to lower porosity and denser compounds. We potentially underestimate the relative contributions of the variable distribution slope, since it only features limited variability in M<sup>4</sup>AGO. In order to better understand and quantify the feedback proposed by Leung et al. (2021), incorporation of ballasting material and potentially associated microstructural changes of particles in their approach would be thus helpful.

In summary, major future changes and uncertainties relevant for sinking fluxes are associated to mean primary particle density, related to ballasting minerals and fecal pellet production, and microstructure of particles. Given that particle fluxes theoretically offer a better constrain on ocean biogeochemistry models than nutrient tracer distributions alone (Kriest et al., 2023), our present study informs about sensitive particle properties that may be changing and affecting fluxes under climate change and thus qualify as present-day observational and process-based modelling targets. Particularly processes that lead to and affect microstructure are highly uncertain while seeming important to understand the response of sinking velocity and thus POC fluxes under a changing climate. Among those constituents and processes that affect particle structure likely most, are the production of sticky extracellular polymeric substances such as transparent exopolymer particles (TEPs; Mari et al., 2017; Quigg et al., 2021), effects of compounds on overall particle aggregation efficiency and susceptibility to fragmentation and re-packing processes (Briggs et al., 2020; Song et al., 2023), but also biologically mediated processes such as packing in fecal

pellets (Kilps et al., 1994) through zooplankton ingestion of phytoplankton or during coprophagie. Initially investigated tediously via SCUBA sampling (Alldredge and Gotschalk, 1988), theoretically (Jiang and Logan, 1991; Logan and Kilps, 1995), or under controlled laboratory conditions (Hamm, 2002; Passow and De La Rocha, 2006), the role of variable composition and microstructure for particle transport (Omand et al., 2020; Trudnowska et al., 2021; Cael et al., 2021), but also for optical properties (Organelli et al., 2018; Anitas, 2020; Wang et al., 2022) relevant for satellite inversions and underwater light climate are increasingly acknowledged. However, little information is thus far available on processes governing particle microstructure in a microbially structured, complex, heterogeneous ocean (Azam and Malfatti, 2007) and how to incorporate these less heuristically and more mechanistically in models like M<sup>4</sup>AGO to reduce the associated uncertainties.

#### 3.5 Evolution of remineralization length scales

In a warming climate, the ocean takes up heat leading to rising sea water temperature. From the present day to the future period, particularly the northern and partially also the southern high latitudes experience a strong warming by regionally more than 100% (Fig. 10a). In the low latitudes, the near-surface waters and the upper mesopelagic also experience rising temperatures of about 1.5 °C to 2 °C. In the M<sup>4</sup>AGO run, these rising temperatures decrease the molecular viscosity of sea water, leading to regional reductions of up to about -10%, which increases settling velocities of particles. A major impact of the rising temperature is on the remineralization rate, which can strongly affect the remineralization length scales of detritus and opal (Fig. 10). The remineralization length scale (RLS; the e-folding depth) for the individual particle component  $X = \{\text{detritus, opal}\}$  is defined by

$$RLS(X) = \frac{\langle w_s \rangle}{\mu_{\text{remin}|\text{dissolution}}} \tag{4}$$

where we stick to the terminology of RLS irrespective of the processes of remineralization (in the case of detritus) or dissolution (in the case of opal). RLS combine the effect of  $\langle w_s \rangle$  and remineralization (or dissolution) and ultimately aggregate effects into a length scale that determines flux attenuation.

Generally, the RLS for opal are much larger than for POC. The faster remineralization of detritus through rising temperature leads to a loss of particles buoyancy and thus to relatively larger RLS of opal, while the increased dissolution of opal can cause less ballasting and even smaller RLS for detritus. The relative changes of dissolution and remineralization of opal and detritus, respectively, due to temperature, determined via the different  $Q_{10}$  factors, also contribute to these linked effect on RLS. In the subtropical gyres, where coccolithophores produce ballasting CaCO<sub>3</sub>, the loss of buoyancy through increasing temperature-driven, faster detritus remineralization and repacking can lead to longer RLS. This is particularly true up to the depth of the lysocline, until which CaCO<sub>3</sub> experiences little dissolution in HAMOCC. Further, dust is handled inert during oceanic transportation and sinking. Thus, in  $M^4AGO$ , the relative contribution of the individual particle components locally co-determine the individual RLS and thus their changes with climate warming while they are closely linked to structural changes in the phytoplankton and zooplankton community structure.

The aforementioned linkages between the individual particle components and export processes lead to a complex response of the RLS (Fig. 10b,c). In the low latitudes, RLS experience an increase, primarily driven through temperature-driven, faster

**Figure 10.** Historical period zonal mean values, absolute and percentage change of the zonal mean value in the future period for a) temperature, b) remineralization length scales (RLS) for detritus and c) RLS for opal in M<sup>4</sup>AGO. Note that the RLS are not weighted by fluxes and thus do not necessarily translate into zonal changes of vertical fluxes of the shown components.

remineralization of detritus in the euphotic zone partially accompanied by an enhanced CaCO<sub>3</sub> to opal export ratio. In the subtropical gyre region, the RLS decrease. In the equatorial region and deep ocean the effect of ODZs on the RLS of POC is well visible. The decreasing remineralization under oxygen limitation increases the RLS leading to the higher transfer efficiency in ODZ regions (see Sec. 3.2). The decline of the POC RLS between 20 ° N to 40 ° N at 1000 m to 2000 m in the





equatorial and northern subtropical region are thus likely linked to changing geometries of and declining deep ODZs. In the northern high latitudes, the Arctic Ocean, NPP increases through sea ice loss and produced detritus adds particle buoyancy leading to strongly declining RLS (Fig. 10).

Coupling of RLS requires deeper understanding of the interaction between minerals and any kind of detritus. Studies by e. g Khelifa and Hill (2006); Sanders et al. (2010) and Spencer et al. (2021) cast doubt on the assumption of statistically well-homogenized particle size distributions and rather suggest certain clusters, in which e.g. minerals prevail in smaller particles, while large, loose particles are dominated by high organic to inorganic ratios, which could have their physical basis in the different physico-chemical surface properties of the components and different timescales of aggregation and intrasize distribution homogenization through e.g. recurring aggregation and fragmentation processes. Changes of RLS in the mesopelgic region can affect resupply and sustaining of nutrients in the euphotic zone in combination with vertical mixing and short term and seasonal excursions of the mixed layer depth (Leung et al., 2021). Overall, the Arctic Ocean presents a hotspot, where RLS and thus transfer efficiency substantially decrease. This contributes to the increased NPP (see Sec. 3.1 and 3.2) and thus poses a positive feedback loop to NPP that is currently not represented in the CMIP6 model version.

#### 3.6 **General Discussion**

The response of net primary production and associated export fluxes from the euphotic into the mesopelagic zone under projected climate warming are thus far afflicted with high uncertainties, and ESMs even disagree in the sign of response (Laufkötter et al., 2015; Henson et al., 2022). This limits the ability to constrain the future biological carbon pump and thus carbon sequestration with vast consequences on e.g. fisheries management decision making and deep ocean diversity management. The response of the net primary production, export production and generally biological carbon pump in ESMs is linked to the incorporated processes and their representation (Laufkötter et al., 2015; Laufkötter et al., 2016). For example, ESMs only partially represent relevant mechanisms for present-day and future export fluxes (Henson et al., 2022). To some extend, this is due to limited observations, unknown constraints and thus the ability to adequately represent the processes. By simplifying or ignoring processes, compensating effects might be excluded and biases are introduced during the model tuning which, as a consequence, induces uncertainties in future projections (Henson et al., 2022). With the present work we therefore aimed at increasing the processes representations in HAMOCC as part of MPI-ESM and the understanding how that affects the future export fluxes and biological carbon pump.

The comparison of the two MPI-ESM1.2-LR model versions, CMIP6 and M<sup>4</sup>AGO, enabled to investigate the response 515 of ocean biogeochemical fluxes and biogeochemistry to two different sinking schemes and remineralization parametrizations under ongoing climate warming. Studying the two sinking schemes with the same physical model, the differences beteen the simulations are attributed to responses to the changed sinking dynamics. The most striking changes associated to the different sinking schemes are related to the change in remineralization length scales in the euphotic zone and upper ocean. Maerz et al. (2020) showed that both, sinking velocity and temperature-dependent remineralization contribute to changes in RLS of POC. M<sup>4</sup>AGOs RLS are particularly shorter in the euphotic zone of the low latitudes than in the standard CMIP6 version. This promotes higher NPP and buffering of its future decline in the low latitudes (Segschneider and Bendtsen, 2013). The






different export flux and transfer efficiency pattern imprints also on the latitudinal  $CO_2$  flux pattern and the timing, when the low latitudes turn into a net-sink of  $CO_2$ . Despite regional differences between the two model versions, the response of the global  $CO_2$  fluxes are comparable with a 0.5% difference in totally taken up  $CO_2$  during the simulated time between 1850 and 2100. Internal variability-elucidating ensemble simulations need to be carried out to further detail effects of transfer efficiency pattern on oceanic carbon sequestration potential.

In addition to an advanced representation of marine particles, M<sup>4</sup>AGO is computationally inexpensive compared to size class-based models (e. g. MSPACMAM; Dinauer et al., 2022) that require one tracer per size class and settling component which, for example, doubles the number of advected tracers for a two size class model. Computation costs are yet a nonnegligible factor in Earth system modelling. The low computational costs and explicit representation of ballasting effects also of dust on particles suggest M<sup>4</sup>AGO also as candidate in paleo studies (Liu et al., 2024), e. g. for understanding dust contributions to atmospheric CO<sub>2</sub> variations during glacial-interglacial cycles. The low computational costs of M<sup>4</sup>AGO and the mechanistic representation of particle dynamics led to successfully applying it in a global ocean biogeochemistry model dedicated to seamlessly integrate a high resolution coastal ocean, ICON-COAST (Mathis et al., 2022, 2024). The new sinking scheme performed reasonable well even under highly dynamic conditions, including benthic-pelagic coupling, typical for coastal and shelf sea regions and was able to reproduce characteristic spatio-temporal properties of particles. These features make an application in high-resolution, sub-mesoscale resolving ocean models possible and promising (Jungclaus et al., 2022; Hohenegger et al., 2023), while the thus far limited representation of size distribution dynamics and internal homogeneous particle composition poses challenges to represent particle dynamics in high resolution models adequately.

In summary, uncertainties and limited process understanding still poses a challenge, but introducing the complexity of particles and their dynamics in ocean biogeochemical models opens new research avenues to investigate the evolution of the future biological carbon pump.

## 4 Conclusions

The development of M<sup>4</sup>AGO offered the possibility to study the response of net primary production, export fluxes and the biological carbon pump to future climate warming via an explicit and thus more realistic representation of marine particles in an ESM. The global effect of M<sup>4</sup>AGO on future NPP, export fluxes and carbon fluxes are comparable to the MPI-ESM1.2-LR CMIP6 version while they both compare well to results of other ESMs of the CMIP6 cohort. Largest differences between the CMIP6 and the M<sup>4</sup>AGO version of MPI-ESM1.2-LR occur at regional scale and their response differ particularly latitudinally. With M<sup>4</sup>AGO, MPI-ESM shows stronger buffering, yet a slightly stronger future decline of NPP in the tropical and subtropical regions due to nitrate loss in more extended ODZs. In the Arctic, the future increase of NPP due to ice-free seasons is weakened in M<sup>4</sup>AGO by its higher transfer efficiency and thus nutrient loss compared to the CMIP6 version. The different latitudinal pattern with M<sup>4</sup>AGO compared to the CMIP6 model version imprints on CO<sub>2</sub> fluxes. Higher latitudes, particularly the Southern Ocean, gain more importance in oceanic CO<sub>2</sub> uptake which also delays the timing of low latitudes becoming a net-sink of CO<sub>2</sub>.



M<sup>4</sup>AGOs mechanistic representation of particles links modeled particle fluxes closer to observations. Despite comprising a number of uncertain parameters, M<sup>4</sup>AGO thus contributes in making research needs transparent, i.e. on microstructure of particles in a heterogeneous ocean which together with composition i) determines settling behavior and ii) turned out to strongly affect future sinking velocities of particles and thus indicating a considerable uncertainty for future fluxes. In addition to microstructure, the temperature-dependent POC remineralization and mineral dissolution co-determine and affect the remineralization length scales and indirectly the vertical evolution of settling velocities in M<sup>4</sup>AGO. Better understanding mineral to POC associations, and potential changes in primary particle sizes will aid in further constraining the ballasting and size effects on the biological carbon pump in a future climate.

In conclusion, at the regional scale the more mechanistic sinking scheme M<sup>4</sup>AGO that responds dynamically to changing environment and primary production, buffers future ocean biogeochemistry more compared to the stiff Martin curve-like POC flux representation.

Code and data availability. MPIOM model code for the M<sup>4</sup>AGO simulation, git sha ID: 32878f1ca68e9ad62e60a5baee688ff69e45694, is available upon request to the author. For The MPI-ESM code and licensing, refer to: Model Development Team Max-Planck-Institut für Meterologie (2024). Main primary data and analysis scripts are stored and made available through the Open Research Data Repository of the Max Planck Society Edmond: Maerz (2025); last access 16.09.2025 (Max Planck Society, Munich, 2020).

#### Appendix A: Injection fraction of large particles at export depth

In order to facilitate comparison to particle flux models like MSPACMAM (Dinauer et al., 2022) that consider small and large particles for sinking fluxes, we here calculate the injection fraction of large particles at export depth  $R_{\text{inj}} = \frac{C_l}{C_s + C_l}$ , where:

$$C_s = \int_{\langle d_{\rm p} \rangle}^{d_{\rm s}} n(d) \, m(d) \, \mathrm{d}d \quad ; \quad C_l = \int_{d_s}^{d_{\rm max}} n(d) \, m(d) \, \mathrm{d}d \tag{A1}$$

with n(d) and m(d) being the number distribution and particle mass, respectively. We here follow Dinauer et al. (2022) and set the splitting diameter to  $d_s = 250 \,\mu\text{m}$ .

We find a similar latitudinal pattern as Dinauer et al. (2022) with higher POC injection to large particles in the high latitudes and upwelling regions (Fig. A1). However, our model shows higher injection fraction to large particles in the subtropical gyres and tropical regions than used in Dinauer et al. (2022). Generally, the high latitudes exhibit a higher variability of the injection ratio due to higher variability in large particle formation in our simulation, visualized as standard deviation of monthly injection fraction.

**Figure A1.** Climatological mean a) and standard deviation b) of the monthly injection fraction of large particles (250 µm; following Dinauer et al., 2022) at export depth for the historical period. c) Injection fraction applied by Dinauer et al. (2022). d) Difference of the injection fraction between M<sup>4</sup>AGO and Dinauer et al. (2022).

#### 580 Appendix B: General Evaluation




On long time scales, circulation in combination with the spatial pattern and amount of export production and attenuation of POM fluxes imprints on nutrient distributions. Particularly the high transfer efficiency in high latitudes and the shallower remineralization in subtropical and equatorial regions in M<sup>4</sup>AGO compared to the CMIP6 version can therefore be expected to lead to changes in nutrient, DIC and alkalinity distributions. Generally, MPI-ESM1.2-LR circulation cannot be expected to fully represent real world circulation in all details, which introduces biases not only in physical variables (Jungclaus et al., 2013; Mauritsen et al., 2019), but also in ocean biogeochemistry. We therefore briefly evaluate the two model versions via Taylor diagrams (Taylor, 2001) on an unweighted grid point comparison basis. Since the grid spacing roughly varies with the Rossby radius of deformation in MPI-ESM1.2-LR, the unweighted grid point-based comparison enables to put similar weights between low- and high latitude regions. In an area-weighted approach, high latitude regions would have lower influence on model biases in our show cases.

Analogue to Maerz et al. (2020), we distinguish between two different ocean regions, the Pacific and the Atlantic ocean, to account for their different circulations and water mass ages and compare them to gridded climatological data products at four different depth (Fig. B1). Overall, the two model versions show very similar biogeochemical tracer distributions and biases compared to observational products, despite their structural differences. There is no clear tendency of improvement or deterioration of M<sup>4</sup>AGO compared to the CMIP6 model. One potential exception with regards to the overall agreement

**Figure B1.** Taylor diagrams for historical period. We compare simulated historical climatological means of silicate, phosphate, oxygen and nitrate distributions to the World Ocean Atlas (WOA) version 13 (Boyer et al., 2013; Garcia et al., 2014a, b). For total alkalinity and DIC, we compare to the Global Ocean Data Analysis Project (GLODAPv2) climatology (Lauvset et al., 2016; Olsen et al., 2016). Note that Si is outside the axis in the euphotic Atlantic, where both models show a correlation of about 0.5, normalized standard deviations of about 5.5/6.5 and 3.0/3.8, and RMSD of 5./6. and 2.6/3.4, for M<sup>4</sup>AGO/CMIP6 at 6 m and 100 m depth respectively.





between the two model versions is silicate, for which the differences between the two model versions peak out and CMIP6 showing higher normalized standard deviations and root mean square deviations (RMSD) than M<sup>4</sup>AGO. It is noteworthy to mention again that the CMIP6 version exhibits a too high silicate inventory, which is not the case in the M<sup>4</sup>AGO version (see Sec. 2.2). These CMIP6 biases of Si concentrations are particularly high in the deep ocean (not shown). We would, however, expect the surface ocean to be most influenced by biogeochemical processes like uptake, sinking and dissolution, which would suggest a structural influence on the model representation of silicate particularly in euphotic waters. Typically, silicate, together with phosphate, are regarded as less influenced by biogeochemical processes, e.g. compared to nitrogen species, and are thus indications for the goodness of ocean circulation. In M<sup>4</sup>AGO, silicate dynamics is stronger linked to biogeochemical processes due to the coupling of settling velocities and remineralization that can take more effect. Further, we observe a too high residence time and thus dissolution in ODZs, which is closely linked to sluggish circulation inherent to general ocean circulation models of resolution classically used in ESMs (Najjar et al., 1992; Aumont et al., 1999; Dietze and Loeptien, 2013; Duteil et al., 2021). Likely, we additionally overestimate opal dissolution in these ODZs, since we do not account for potential oxygen limitations of the microbially mediated dissolution rates (Bidle et al., 2002). Generally, the overestimated ODZs impinge on nutrient, DIC and alkalinity biases in deeper ocean regions.

# 610 Appendix C: Future changes in mineral export fluxes in M<sup>4</sup>AGO

Maerz et al. (2020) showed that mean sinking velocity of particle mass concentration is strongly linked to particle properties, among them most influential primary particle density, size and particle microstructure, which in M<sup>4</sup>AGO are all determined by settling tracer concentrations and their ratios. In the following, we therefore showcase POC and biogenic mineral fluxes, their ratios and their changes, which affect particle properties in future (see Sec. 3.4). We here focus on export fluxes at the euphotic depth, since they also determine to great extend, how sinking particles and their properties populate through the water column and thus determine future changes of sinking velocity (Sec. 3.4), and the attenuation of fluxes reflected by remineralization length scales (Sec. 3.5)

The general pattern of POC export fluxes,  $F_{POC}$ , follows, as expected, the net primary production (cmp. Fig. C1 and Fig. 1). Highest  $F_{POC}$  occur in nutrient-rich upwelling regions and in the equatorial eastern Pacific, where equatorial jets sustain mixing and thus high nutrient delivery into the euphotic zone. Significant opal production and subsequent sinking and export to the mesopelagic is associated and confined to these upwelling as well as well-mixed or seasonally less stratified regions. The close connection between POC and opal production through phytoplankton growth and decay in HAMOCC leads to an opal to POC ratio pattern at export depth with lower ratios in equatorial and subtropical regions than in high latitude regions. This latitudinal pattern emerges since POM remineralization is generally faster than opal dissolution and POM remineralization is strongly enhanced in the warmer regions. The latitudinal pattern is closely linked to the higher temperature-dependent POM remineralization leading to higher POM losses compared to opal. Inherent to HAMOCC, once bulk phytoplankton and zooplankton become silicate-limited, organisms are assumed to start producing shell material via calcification which is mainly confined to lower latitudes. Highest CaCO<sub>3</sub> export fluxes are found in the westward wake of the opal production region in




Figure C1. Historical fluxes of POC ( $F_{POC}$ ), CaCO3 ( $F_{CaCO_3}$ ) and opal ( $F_{opal}$ ) at 100 m, their ratios and changes in M<sup>4</sup>AGO from historical period to end of the century.

the equatorial jet of the Pacific. Here, nutrient availability is still high through remineralization, while silicate is diminished due to opal production, export and slow dissolution. High remineralization rates, non-shell producing plankton including cyanobacteria shift the highest CaCO<sub>3</sub> to POC export ratios further westward and into the southern subtropical gyre. The rather sequentially process order of silification and calcification, dependent on silicate availability, leads to a distinct latitudinal opal to CaCO<sub>3</sub> export ratio with high values in opal-dominated regions.

In line with changes of the *p* ratio and NPP discussed in Sec. 3.1, future POC fluxes decline in most parts of the subtropical and tropical regions. An exception is associated to the recirculation of the Humboldt current, where increased POC export production are found in the projection, similarly in CMIP6 (not shown). In higher latitudes, increased fluxes appear in the Subantarctic frontal region and in the Arctic Ocean. A similar pattern of opal export fluxes appear. By contrast, the opal to POC flux ratio shows a bipolar pattern in northern and southern high latitudes. While in the silicate-rich Southern Ocean, the ratio marginally changes, the ratio declines in the northern Pacific and Arctic Ocean. In the case of CaCO<sub>3</sub> fluxes, the northern Pacific, North Atlantic and the subtropical gyres show increasing CaCO<sub>3</sub> export production in the future period. The CaCO<sub>3</sub> to POC export ratio increases strongly in most subtropical and tropical regions, but also in the northern higher latitudes. The formerly described bipolar future response in northern and southern high latitudes for opal export fluxes manifests also in





the opal to  $CaCO_3$  export production ratio.  $CaCO_3$  gains higher relevance compared to opal export production in the future northern latitudes according to  $M^4AGO$ .

For most ocean regions, biogenic mineral (opal and CaCO<sub>3</sub>) relative to POC export increases in future and thus provides relative more ballasting (including primary particle size effects; Maerz et al., 2020) to particles. The future decrease in POC to mineral ballast ratio at export depth can primarily be attributed to increased remineralization due to rising surface ocean temperatures. The changes in opal and CaCO<sub>3</sub> export production are a consequence of declining NPP and changing availability of silicate. The latter is primarily associated to changing stratification. In regions of future intensifying stratification, silicate lost early in the growth season cannot be mixed up easily again into the euphotic zone to sustain or promote further silicification. By contrast, the continuous strong mixing in the Southern Ocean is sustaining the ongoing opal production-dominated phytoplankton community.

To briefly summarize, the interplay of changes due to temperature-dependent remineralization, NPP and stratification-related dissolved silicate availability imprint on export fluxes and their ratios which also manifest in particle property changes, discussed in Sec. 3.4.

## Appendix D: Simulated particle sinking velocity and their future changes in M<sup>4</sup>AGO

In Fig. D1, the climatological mean sinking velocity for the historical period and the respective changes for the future period are shown for  $M^4AGO$  for the euphotic depth (100 m).

Figure D1. Left: Climatological mean of the concentration-weighted mean sinking velocity at 100 m depth for the historical period. Right: Changes of  $\langle w_s \rangle$  in the future period.

Author contributions. Conceptualization: JM, KDS and TI conceptualized the research goals. Data curation: KDS and JM took care of the simulations. Formal analysis: JM carried out the analysis of model results. Funding acquisition: TI wrote and acted as PI of the initial MARMA proposal. JM and SA wrote the follow-up proposal for MARMA. TI received the CLICCS project grant and thanks to Christoph Heinze, who got granted the ESM2025 funding under which JM is working; JM received funding through BioGeoSea. Investigation: JM conducted the experiments and research with support by KDS and Irene Stemmler. Methodology: JM earlier developed the advanced


sinking scheme. **Project administration:** TI had the responsibility for the project administration. **Resources:** The DKRZ provided the HPC resources, where all computations and post-processing were carried out. **Software:** The earlier implementation of the new sinking scheme was performed by JM, with technical support and code review by Irene Stemmler and KDS. **Supervision:** No supervision involved. **Validation:** Evaluation of the model was performed by JM. **Visualization:** Visualization of model experiments was carried out by JM. **Writing – original draft preparation:** JM prepared the original draft of the manuscript. **Writing – review & editing:** KDS and SA contributed significantly by reviewing the manuscript. JM performed the editing.

Competing interests. The authors declare no competing interests.

Acknowledgements. The authors thank Hongmei Li for the internal review and comments on the manuscript and Irene Stemmler for technical and scientific support of the work. The Max Planck Society (MPG) funded the project 'Multiscale Approach on the Role of Marine Aggregates (MARMA)' within which a dominant part of the work was carried out. Further, the Deutsche Forschungsgemeinschaft (Germany's Excellence Strategy – EXC 2037 "CLICCS – Climate, Climatic Change, and Society" – project no. 390683824, contribution to the Center for Earth System Research and Sustainability (CEN) of Universität Hamburg) and the European Commission H2020 (Earth system models for the future - ESM2025, grant no. 101003536; and BioGeoSea, grant no. 101216427) contributed to financing this work. SA thanks the Leibniz-Association (strategic institute expansion: "Shallow Water Processes and Transitions to the Baltic Scale"), Novo Nordisk Foundation (Grant No. 0079370). All simulations and post-processing were carried out at the German Climate Computing Center (DKRZ).

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
