# Peer review of "Marine particles and their remineralization buffer future ocean biogeochemistry response to climate warming"

_EGUsphere, 2025_

## Author Comment (AC1)

**1 Responses to Reviewer I**

Dear Reviewer,

many thanks for you valuable comments on our manuscript. In the document below, please find

⇒ in blue your initial review comment

⇒ in darkcyan our reply, with potential indicated  → modifications/additions in the manuscript text

Kind regards,

   Joeran Maerz on behalf of the co-authors

**1.1 General comment**

The study by Maerz et al. provide an extensive analysis of the M4AGO parametrisation in a context of climate change. This parametrization includes temperature-dependant remineralization, oxygen limitation of remineralization, sea water viscosity, ballasting (composition) and a microstructure (fractal dimension / porosity) representation with aggregation/desagregation processes including particle density, size and stickiness. Sinking velocity is ultimately considering sea water viscositiy, particle composition (of ballast material), particle density, porosity and size. The study is very well written (although very verbose and some times convoluted), and provide a dense, comprehensive, well referenced, honest (especially about the limited impacts on global air-sea CO2 fluxes and limitations in general) and transparent analysis of this ambitious parametrization. The authors demonstrate a very high level of mastery in their disciplines.

⇒ Many thanks. We very much value the appreciative tone of the review.

They found little influence and global scale but highlighted regional important differences such as in the Arctic Ocean.

The review of this article was challenging. About 23 pages that must also include the lengthy study of Maerz et al. 2020 (another very lengthy and technical paper introducing the M4AGO parametrization). The writing is sometimes lengthy and technical. The paper in general would deserve a more synthetic and accessible bite. The problem with that is I really wonder who is able to read and actually digest this article beyond the small BGC modeling community.

⇒ We acknowledge that reviewing such manuscript is demanding, in parts due to its reliance on the previous publication of Maerz et al. (2020) which introduces M$^4$AGO. When revising the manuscript, we will sharpen the presentation to improve accessibility while retaining the technical detail required for the targeted biogeochemical modeling and observational communities.

The other problem is related to the microstructure parametrization which the authors claim is an important factor elucidating regional patterns of the BCP. If all other parametrizations are relatively simple and are developed similarly in other models, the microstructure parametrization increase complexity substantially with a lot of under- (or non-) -constrained parameters (see Maerz et al. 2020). I am aware that the authors already acknowledge this, guaranty computational efficiency and provide quantitative effects. Still, this is very hard to proof what is done here especially noting that the code is not open. How can this parametrization could be evaluated with observations? (The comparison with CO2 fluxes does not necessarily show an improvement to be honest). How could we constrain more the numerous parameters? (although if not achievable now, what could be used in the future?) I know they acknowledge there is little information available so far, but how could we proceed then? Are we sure the regional patterns are more realistic?

⇒ Maerz et al. (2020) aimed at very thoroughly documenting M$^4$AGO including its underlying assumptions and potential weaknesses to make it accessible to other researchers. This is supported by the code availability being open as part of the current MPIOM master branch (the ocean component of MPI-ESM) as well as ICON-O, the successor model, both licensed officially under the BSD-3-C License. In addition, the first author is the maintainer of a publicly available M$^4$AGO standalone code basis, which is under active further development, also licensed under BSD-3-C License. We consider to provide this information on the repository explicitly as well.

As already partially demonstrated by Maerz et al. (2020), M$^4$AGO can be regarded and serve as an extendable framework that aims to bring observations and models closer to each other. We acknowledge that some parameter uncertainties in M$^4$AGO are high. However, in contrast to simpler parametrizations, all parameters of M$^4$AGO are technically measurable and are consistently and mechanistically linked to each other. Thus far, limited observational throughput and thus

statistics certainly limit the capability for comparison. For example, $M^4AGO$ rather simulates mean particle than individual particle properties. However, (subsets of) the methods underlying $M^4AGO$ are applicable even for individual particles and thus could build a bridge from individual particles to $M^4AGO$ which would allow to evaluate the model approach. In terms of regional patterns, our comparison of e.g. the transfer efficiency through climatological fluxes as derived by Weber et al. (2016) likely provides a reasonable constrain at least for present day climate which we better capture with $M^4AGO$ (as previously discussed in Maerz et al., 2020). We are, however, fully aware that $M^4AGO$ has its limitations, which we discussed in Maerz et al. (2020) and the present manuscript. Nevertheless, we are optimistic that $M^4AGO$ captures general particle features (see also comment on line 410ff below).

Since $CO_2$ fluxes, as discussed in the manuscript, are strongly governed by circulation, $CO_2$ not necessarily provides the best measure to judge advancements in ocean biogeochemistry. Given the deficiencies in representing ocean circulation and ocean mixing, we believe that ocean biogeochemistry modelling is a bit stuck in a 'performance-limbo' - trying to build -easily- tunable OBGC models that tend to cover ocean circulation deficiencies (e.g. limited -sub-mesoscale- eddy representations, sluggish circulations responsible for extended oxygen deficit zones as discussed in the manuscript, etc.). Circulation deficiencies tend to be become amplified in OBGC model responses. As a consequence, OBGC models tend to lack fidelity in process representation (e.g. in our case the Martin-curve like representation, where e.g. transfer efficiency is almost fixed entirely) and response to climate change in future scenarios. The latter underpins the need to explore a variety of model representations, i.e. here via $M^4AGO$ for the biological carbon pump. Bringing datasets together or building new, more complete datasets would thus be beneficial - i.e. featuring particle composition, size, density and sinking velocity, ideally including further environmental variables like temperature, salinity, turbulent shear rate and (vertical) turbulent mixing rates. We further believe that with increasingly available optical data, even the link between particle composition (based on particle grey scale values and/or differences in RGB channels), particle density and size could be established in a statistically meaningful sense, which would enable to verify or falsify $M^4AGO$'s underlying assumptions and to expand $M^4AGO$ further.

On the regional aspect, the authors put a lot of emphasis on the Arctic

Ocean (and OMZ). If conclusions rather make sense for most regions to me, I was still puzzled by the conclusions drawn for the Arctic mostly because of the lack of synthesis capacities of the authors. Explanations are scattered around which makes a lot of work for the reader to re-assemble the results and conclusions. They show that M4AGO allows higher transfer efficiency compared with CMIP6 (Appendix D show maximum sinking velocity in the Arctic? Why?). But climate induce change towards:

– More NPP, more export, less compact & larger particles, more buoyant (decrease sinking velocity)

– Warming temperatures decrease viscosity (increase sinking velocity)

– Less Opal / more calcite compared to POC in the future (total effect?)

$\rightarrow$ If I understand well, this overall has the effect of decreasing the sinking velocity in the Arctic (Appendix D)

However, this is combined with:

Warming temperatures increase remineralization

I finally got the sense of the overall message: The total effect is RLS & transfer efficiency decrease despite increase in NPP (positive feedback loop). The authors should wrap this up somewhere better, it's not an intuitive result. Same for other regions eventually.

$\Rightarrow$ Many thanks. We will provide an aggregated view on the underlying responses by showing a radar plot (Fig. 1). For the revision, we will likely further split the current graphics up (and consider aggregating it further) and focus on a subset of the ocean regions (e.g. the Arctic Ocean region), while showing the remaining ocean regions in the appendix.
For further explanation on this matter, in the Martin-curve approach in the CMIP6 version, the annual mean transfer efficiency remains fairly fixed for both periods that is smaller than in $M^4AGO$ in the high latitude regions (see Fig. 4 of the manuscript; the fixed transfer efficiency also becoming immediately clear from Eq. (34) in Maerz et al., 2020, when considering globally fixed remineralization rates and linearly increasing sinking velocity with depth that determines the Martin curve slope - note that this is only for climatological POC fluxes, while on

sub-annual time scales, the transfer efficiency can still vary a lot due to time lags in vertical POC fluxes with depth). This in itself already has an effect on the different response particularly in the Arctic Ocean (i.e. higher transfer efficiency can be associated with deeper nutrient loss and thus less NPP). Additionally in $M^4$AGO, the increased NPP in the Arctic Ocean leads to more buoyant particles (decreasing fractal dimension and mean primary particle density, while increasing mean primary particle size and maximum particle size). As an overall effect, this reduces the RLS (see also Fig. 3 below in this reply) and thus reduces transfer efficiency in the future (while it still remains higher than the transfer efficiency modeled via the Martin-curve approach).

[Figure]

Figure 1: Radar plot for climatological percentage difference between the projection and the historical time period of monthly, area-weighted mean particle properties, particulate export fluxes and molecular dynamic viscosity at export depth for major ocean regions. Note that first row represent different ±-percentage changes than the rest of the subplots.

This article is certainly worth publishing, but I would recommend a few changes and clearer explanations before doing so.

**1.2  Specific comments**

I have noted point-by-point comments below:

Line 107: What are the limitations of such hypothesis? In general this does not stand in case of strong lateral advection.

⇒ We agree with the general limitations through a fixed power law form of a size distribution representation. However, the power law size distribution is widely used and accepted as reasonable approximation in literature on open ocean particle size spectra. We discuss the limitations of the fixed size distribution form in the general discussion section (and discussed it also previously in Maerz et al., 2020) particularly for high resolution applications (l. 538ff of the initial submission). Note, however, that these limitations cannot be overcome by more simplified model representations, either. To mention it, we were positively surprised to see i.e. the spatial structure of sinking velocity and other particle properties in a cross-shore transect in the ICON coast setup that is well in line with observationally based findings (cmp. Fig. 12 in Mathis et al. (2022) to Fig. 4 and 5 in Maerz et al. (2016)). This is particularly possible, since M$^4$AGO captures both, size distribution and particle density, in a physically reasonable manner which is a step forward in bringing these key particle properties explicitly together. We are, however, aware that it ideally deserves a thorough testing where the parametrization breaks or produces too large biases and that it deserves future investigations on how to incorporate the represented particle properties more adequately in models, when also better capturing size distribution dynamics. Both, further testing and incorporating size dynamics while remaining computationally feasible, remain challenging. We address the comment by more explicitly mentioning the fixed functional form issue in the general discussion section: "These features make an application in high-resolution, sub-mesoscale resolving ocean models possible and promising (Jungclaus et al., 2022; Hohenegger et al., 2023; Nielsen et al., 2025), while the thus far limited representation of size distribution dynamics → , namely through a fixed functional form and lower represented variability than measured, and internal homogeneous particle composition poses challenges to represent particle dynamics in high resolution models adequately."

**Section 2.2:** Why not adjusting calcite?

⇒ As a sinking tracer with a water residence time of about a few month, e.g. assuming an average sinking velocity of about $30\,\mathrm{m\,d^{-1}}$, there might be an initial mismatch between adjusted alkalinity and DIC, but this does not affect the long term behavior of the model and leaves minimal trace in the sediment and thus bottom waters. For $CaCO_3$ production, the amount of deposited $CaCO_3$ aligns well with literature values. Hence, we were neglecting performing further adjustments to the $M^4AGO$ model.

**Line 161:** physical internal variability is not assessed, is there any differences in the physical fields?

⇒ The physical model MPIOM is identical in both simulations, $M^4AGO$ and the CMIP6 version. As described, we branched off the physical restart file from the pre-industrial control run of the CMIP6 version. To allow for the adjustment of the biogeochemistry, the spin-up of $M^4AGO$-MPIOM HAMOCC was extended by approximately 700 years under pre-industrial conditions. Please note, that the changes in the biogeochemistry do not feedback to the physical ocean. Therefore, the physical ocean in $M^4AGO$ shows the same internal variability as the CMIP6 version. With respect to the physical fields, the $M^4AGO$ simulation could be regarded as an additional realisation to the 10 existing ensemble members of MPI-ESM1.2-LR CMIP6 simulations available on the ESGF repository.

**Line 165:** Not true. Stratification only decrease in the Atlantic sector of the Arctic. Fix also statement line 227.

⇒ To provide evidence for MPI-ESM showing less stratification during summer in the future, we here provide the monthly evolution and changes of the vertical maximum stratification of the topmost $500\,\mathrm{m}$ in the Arctic Ocean. Changes show clearly a mean weakening particularly towards end of summer in vast areas of the Arctic Ocean for the future in MPI-ESM (true for both simulations, showing very similar response, here shown for the $M^4AGO$ run, see Fig. 2). For the manuscript revision, we consider aggregating the plot further to only show the mean summer season changes.
CMIP6 models represent the Arctic Ocean atlantification and its general response to climate change in a diverging manner (Muilwijk et al., 2023) and annual mean values cannot capture such seasonality-governed

monthly climatological mean in the Arctic Ocean. We will clarify this
further in the manuscript text.

[Figure]

Figure 2: Monthly climatological vertical maximum squared buoyancy frequency, $N^2$, in the topmost $500\,\mathrm{m}$ and its future changes in the Arctic Ocean - here for the M⁴AGO simulation, the CMIP6 simulation shows very similar pattern and response.

Line 174: With all due respect, this sentence is too complicated. There is sea ice now and there always will be ... in winter. You are talking about summer sea ice. Seasonality, I guess you refer to the winter polar night (absence of light → no NPP). And yes the Arctic is a small ocean but what the point if you discuss relative changes in?

⇒ We will rephrase and shorten the sentence as follows: → In the northern latitudes (>66°N), NPP is generally low due to the ice cover at present day, and general seasonality. With continuing warming of the Artic Ocean, NPP increases by 20.3 % in the M$^4$AGO run and with 26.9 % even more in the CMIP 6 run in the future period.

Line 176: 100m is not the euphotic depth. It is a simplified threshold depth considered as the euphotic depth. Of course, much less accurate that an actual calculation of the euphotic depth (variable in time and space) to derive the export production. It's fine! But reformulate.

⇒ We fully agree that the terminology of euphotic depth is debatable in this context. We will redefine it in the model description and will consider 100 m as model-defined export depth in the revised manuscript throughout the text.

Line 179: While still using the SSP585 while we know this is not the way to go? Hausfather, Z. & Peters, G. P. Emissions – the 'business as usual' story is misleading. Nature 577, 618–620 (2020).

⇒ We acknowledge that SSP5-8.5 might overestimate $CO_2$ emissions (both, short and longterm). Our aim with the manuscript was to describe potential responses, which makes the SSP5-8.5 scenario as extreme scenario well suitable. We address the reviewers comment by explicitly mentioning that the SSP5-8.5 scenario is likely overestimating $CO_2$ emissions in Sec. 2.2 line 147: → ... to showcase responses under an extreme scenario, while acknowledging that it likely overestimates $CO_2$ emissions (Hausfather and Peters, 2020).

Line 204: Did I miss the obvious or the remineralization is not shown?

⇒ We initially neglected to show remineralization rates explicitly in this manuscript, since it was discussed and presented in Maerz et al. (2020), for example, see their Fig. 9. With the revision of the present manuscript, we will provide used sinking velocity and remineralization rate for the Martin case explicitly in Sec. 2.1: "Brief model description" (in the

[Figure]

Figure 3: Climatological mean for the historical period and their changes in the future period in $M^4AGO$ for a,b) sinking velocity; c,d) $Q_{10}$-dependent remineralization rate; e,f) remineralization length scales of POM. For comparison, the CMIP6 version features a globally constant sinking velocity of $3.5\,\mathrm{m\,d^{-1}}$ between $0\,\mathrm{m}$ to $100\,\mathrm{m}$ depth, a remineralization rate of $0.026\,\mathrm{d^{-1}}$ (times oxygen limitation) and thus a RLS(POC)$\approx$135 m at export depth (assuming no oxygen limitation here for simplicity).

text and in a table, see also comments for reviewer II). Further, we will additionally to the sinking velocities and their changes in $M^4AGO$ provide the remineralization rates and their changes and the remineralization length scales - all for the $100\,\mathrm{m}$ export depth - in the appendix Fig. D1. See here Fig. 3

**Line 233:** Sequestration. I have also used this word wrongly for while, I am not blaming, but could we fix that? You can refer to the nice Visser 2025 which clarifies: "carbon sequestration is synonymous with an offset of carbon emissions" https://doi.org/10.1002/lol2.70053 replace by storage at greater depth or similar.

⇒ As far as we understand, Visser (2025) aims at advocating for applying the term sequestration only for the offset of anthrophogenic emissions. We agree that clarity in the terminology - also congruent with the definition in the IPCC report - is desirable and we will review our manuscript and follow the advice of the reviewer, where applicable. In line 233, we perform the modification:  → storage

**Line 255:** Arctic Ocean amplification, ref: Shu, Q. et al. Arctic Ocean amplification in a warming climate in CMIP6 models. Sci. Adv. 8, eabn9755 (2022).

⇒ We will add: In combination with temperature-enhanced remineralization → ... due to Arctic Ocean amplification (Shu et al., 2022).

I agree but this is counter-intuitive for most reader and non-experts. Can you clarify here quickly what is meant? You mean that there is more POM and therefore, relatively, less ballast material in the composition of particles if I refer to Appendix C. Why seasonal average?

⇒ Seasonal was misleading (trying to refer to seasonally varing POC-to-mineral ballast, on average in favor of more POM, which rather complicated the message to convey). We rephrase the sentence to: The additional POM → compared to ballasting minerals increases the buoyancy of marine particles on  → average and thus decreases settling velocity.

**Line 260:** If the inter-annual variability is represented by the STD, say it.

⇒ We will rephrase the sentence to: The largest interannual variability of transfer efficiency → , expressed as interannual standard deviation, is associated to ...

**Line 270-272:** needed?

⇒ Yes, since we would highly advocate for aiming for 1 yr (or more) measurements of the transfer efficiency in ocean regions, which would enable linking observations closer to models and also constrain models

(as laid out by Kriest et al., 2023, for POC fluxes). Short term transfer efficiency calculations suffer from too large dependency on potentially vertically time-lagged fluxes (e. g. Giering et al., 2017, see also Maerz et al. 2020, Fig. C1), which could be overcome by year-long integrated measurements (under the assumption of relatively small inter-annual variability, see Fig. 4 d for regions more/less suitable for such an approach). Ideally this should be accompanied by measurements on processes affecting the RLS over the water column to enable bridging between observations and models. However, we are aware of the high logistical and costly requirements for such observational endeavor, which likely renders it challenging to achieve this. We will consider to move this part in a more aggregated form into the conclusion section.

Line 274: Even a flux cannot! Only change in storage.. See article by Frenger, I. et al. Misconceptions of the marine biological carbon pump in a changing climate: thinking outside the 'export' box. Glob. Change Biol. 30, e17124 (2024).

⇒ We here disagree with the reviewer, since net carbon fluxes (including biological carbon pump and the circulation- and mixing-driven DIC counter pump) ultimately set the storage. Both, Wilson et al. (2022) and Frenger et al. (2024, see their supplementary Figure S2-c) show that the carbon storage is affected by deep ocean carbon fluxes.

Line 309: you mean vertical DIC gradient right? fix through the text.

⇒ Yes. We will precision the respective text.

Line 316: I can understand why (simulations from data product or your simulation) internal variability is a problem, but why the mean of the observational product is?

⇒ Local, time point-wise means of the data product still feature an internal variability over the 30 year time period. This internal variability is not necessarily in phase with the internal variability of the simulations, which complicates the comparison. We will clarify this in the revision.

Line 355: time-cumulative?? you mean yearly integrated?

⇒ Yes, it is time-cumulative (as in time-integrated by summing up). To enable the reader to follow more easily, we will refer to the figure subplot (i.e. current Fig. 8 e,f).

**Line 365:** It is appreciated that the authors acknowledge that physico-chemical process dominate air-sea CO2 fluxes dynamics. Although this is repeated several times in the manuscript.

⇒ While this fact cannot be overemphasized, we agree that the statement is made a few times across the manuscript. We will thus shorten the text and delete the statement in applicable places.

**Line 370:** Yes the BCP if responsible for the most part of the vertical DIC gradient. Rephrase.

⇒ We will rephrase the sentence.

**Line 410:** I don't understand how more detritus production necessarily leads to less compact & bigger particles.

⇒ In $M^4$AGO, detritus features a slightly higher stickiness, leading to a lower fractal dimension and is generally less dense than mineral primary particles. This leads to lower sinking velocities at the same size as e.g. for highly compact, mineral-rich particles. Hence, POM-rich particles can grow larger until the critical particle Reynolds number for fragmentation is reached. From the observational perspective, the authors draw on e.g. coastal turbulent water studies, where the seasonality in particle/floc characteristics is often associated to the availability of fresh organic matter in addition to sediment (mineral) particles. Studies of e.g. Chen et al. (2005) and more recent studies of e.g. Fettweis et al. (2014) and others show frequently that organic-rich particles grow larger and are resistant to fragmentation beyond the Kolmogrorov microscale (a feature that also has been shown for open ocean regions by Takeuchi et al., 2019). In their laboratory study, Hamm (2002) also showed that (sedimentary) mineral ballasting acts as a size-reducing ingredient in a shear-free (apart from potential chamber wall-particle interactions) rolling chamber experiment. For the open ocean, slower sinking and more sticky organic-rich particles have a longer residence time in the upper water column, which also enables them to grow larger due to aggregation processes, before escaping via sinking to deeper regions where no or only little new organic matter for further aggregation is formed.
We will extend this part by providing more information, since it was also recommended by reviewer II to provide more explanation on particle properties in Sec.3.4 to understand particularly the Arctic Ocean

change in transfer efficiency. We extend/modify the text in the following:  → In the Arctic Ocean, where a transition from frequently ice-covered or at least ice-influenced to seasonally ice-free happens in future, increased NPP (see Sec. 3.1), and thus detritus production, leads to, on average, looser, less compact, larger particles. → As modeled by $M^4$AGO, detritus tends to be more sticky than mineral particles, which loosens the internal microstructure of particles due to less intrusion into each other upon collision. Further, the low density of POM compared to minerals lets particles grow larger until reaching the critical particle Reynolds number for fragmentation. While the overall contribution to sinking velocity was ambiguous in laboratory rolling chamber experiments of Hamm (2002), they also found a decrease in particle density and increase in size with decreasing abiotic mineral particle concentration similarly as represented by $M^4$AGO. In $M^4$AGO, this leads to a decrease in sinking velocity in the Arctic Ocean. → By contrast, the increase of water temperature in most parts of the euphotic zone and upper mesopelagic of the oceans leads to lower dynamic sea water viscosity and thus contributes to increasing sinking velocities.

Line 420: Explain me how temperature dependant remineralization has a direct effect on particles density and porosity? You mean temperature in general? I don't understand this sentence.

⇒ We will rephrase the sentence to:  → Enhanced remineralization → in future due to temperature-dependency has thus a two fold effect on particles in $M^4$AGO → in these regions: i) an increase in primary particle density and ii) decreasing porosity, which together lead to denser, faster sinking particles.

As brief explanation here: Increasing temperature increases the remineralization rate (through the $Q_{10}$-approach). Hence, POC is remineralized faster in warmer regions and under future warming. This reduces the mean stickiness of primary particles making the particles more prone for compaction (i.e. stronger intrusion into each other during aggregation, but also less resistant against particle restructuring). This is mimicked by a higher fractal dimension. The process of POC remineralization is faster than the dissolution of opal and $CaCO_3$. As a consequence, particles become less buoyant and more compacted, both of which increases sinking velocity. However, increasing sinking velocity increases the particle Reynolds number and hence, the maximum

diameter at which particles become vulnerable to fragmentation according to our critical particle Reynolds number approach is becoming smaller.

Line 338: variable distribution slope? you mean the size distribution? Not clear to me.

⇒ Yes, we meant the size distribution slope - we will clarify it accordingly, when revising the manuscript by adding → size distribution slope.

Line 500: Likely true. Positive feedback loop maybe see Oziel et al. 2025. Not represented in CMIP6 models... not so sure, prove it.

⇒ We here referred to our CMIP6 model version. We try to make this clearer and refrain to prove it for any other CMIP6-partaking model, which is beyond the scope of the manuscript. For our CMIP6 model version, the feedback loop is not represented.

Line 516: between

⇒ Thanks - we will fix the typo.

Line 545: "more realistic" in terms of process maybe, but in terms of model performance? Not sure.

⇒ Thanks. Yes, we meant process realism, which we will precision when revising the manuscript.

**References**

Chen, M. S., Wartel, S., and Temmerman, S.: Seasonal variation of floc characteristics on tidal flats, the Scheldt estuary, Hydrobiologia, 540, 181–195, 2005.

Fettweis, M., Baeye, M., Van der Zande, D., Van den Eynde, D., and Lee, B. J.: Seasonality of floc strength in the southern North Sea, Journal of Geophysical Research: Oceans, 119, 1911–1926, https://doi.org/10.1002/2013JC009750, 2014.

Frenger, I., Landolfi, A., Kvale, K., Somes, C. J., Oschlies, A., Yao, W., and Koeve, W.: Misconceptions of the marine biological carbon pump in a changing climate: Thinking outside the "export" box, Global Change Biology, p. 30:e17124, https://doi.org/10.1111/gcb.17124, 2024.

Giering, S. L. C., Sanders, R., Martin, A. P., Henson, S. A., Riley, J. S., Marsay, C. M., and Johns, D. G.: Particle flux in the oceans: Callenging the steady state assumption, Global Biogeochemical Cycles, 31, 159–171, 2017.

Hamm, C. E.: Interactive aggregation and sedimentation of diatoms and clay-sized lithogenic material, Limnol. Oceanogr., 47, 1790–1795, 2002.

Hausfather, Z. and Peters, G. P.: Emissions – the 'business as usual' story is misleading, Nature, 577, 618–620, 2020.

Hohenegger, C., Korn, P., Linardakis, L., Redler, R., Schnur, R., Adamidis, P., Bao, J., Bastin, S., Behravesh, M., Bergemann, M., Biercamp, J., Bockelmann, H., Brokopf, R., Brüggemann, N., Casaroli, L., Chegini, F., Datseris, G., Esch, M., George, G., Giorgetta, M., Gutjahr, O., Haak, H., Hanke, M., Ilyina, T., Jahns, T., Jungclaus, J., Kern, M., Klocke, D., Kluft, L., Kölling, T., Kornblueh, L., Kosukhin, S., Kroll, C., Lee, J., Mauritsen, T., Mehlmann, C., Mieslinger, T., Naumann, A. K., Paccini, L., Peinado, A., Praturi, D. S., Putrasahan, D., Rast, S., Riddick, T., Roeber, N., Schmidt, H., Schulzweida, U., Schütte, F., Segura, H., Shevchenko, R., Singh, V., Specht, M., Stephan, C. C., von Storch, J.-S., Vogel, R., Wengel, C., Winkler, M., Ziemen, F., Marotzke, J., and Stevens, B.: ICON-Sapphire: simulating the components of the Earth system and their interactions at kilometer and subkilometer scales, Geoscientific Model Development, 16, 779–811, https://doi.org/10.5194/gmd-16-779-2023, 2023.

Jungclaus, J. H., Lorenz, S. J., Schmidt, H., Brovkin, V., Brüggemann, N., Chegini, F., Crüger, T., De-Vrese, P., Gayler, V., Giorgetta, M. A., Gutjahr, O., Haak, H., Hagemann, S., Hanke, M., Ilyina, T., Korn, P., Kröger, J., Linardakis, L., Mehlmann, C., Mikolajewicz, U., Müller, W. A., Nabel, J. E. M. S., Notz, D., Pohlmann, H., Putrasahan, D. A., Raddatz, T., Ramme, L., Redler, R., Reick, C. H., Riddick, T., Sam, T., Schneck, R., Schnur, R., Schupfner, M., von Storch, J.-S., Wachsmann, F., Wieners, K.-H., Ziemen, F., Stevens, B., Marotzke, J., and Claussen, M.: The ICON Earth System Model Version 1.0, Journal of Advances in Modeling Earth Systems, 14, e2021MS002 813, https://doi.org/10.1029/2021MS002813, e2021MS002813 2021MS002813, 2022.

Kriest, I., Getzlaff, J., Landolfi, A., Sauerland, V., Schartau, M., and Oschlies, A.: Exploring the role of different data types and timescales in the quality of marine biogeochemical model calibration, Biogeosciences, 20, 2645–2669, 2023.

Maerz, J., Hofmeister, R., van der Lee, E. M., Riethmüller, R., and Wirtz, K. W.: Maximum sinking velocities of suspended particulate matter in a coastal transition zone, Biogeosciences, 13, 4863–4876, 2016.

Maerz, J., Six, K. D., Stemmler, I., Ahmerkamp, S., and Ilyina, T.: Microstructure and composition of marine aggregates as co-determinants for vertical particulate organic matter transfer in the global ocean, Biogeosciences, 17, 1765–1803, 2020.

Mathis, M., Logemann, K., Maerz, J., Lacroix, F., Hagemann, S., Chegini, F., Ramme, L., Ilyina, T., Korn, P., and Schrum, C.: Seamless Integration of the coastal Ocean in Global Marine Carbon Cycle Modeling, Journal of Advances in Modeling Earth Systems, 14, e2021MS002 789, https://doi.org/10.1029/2021MS002789, 2022.

Muilwijk, M., Nummelin, A., Heuzé, C., Polyakov, I. V., Zanowski, H., and Smedsrud, L. H.: Divergence in Climate Model Projections of Future Arctic Atlantification, Journal of Climate, 36, 1727–1748, https://doi.org/10.1175/JCLI-D-22-0349.1, 2023.

Nielsen, D. M., Chegini, F., Serra, N., Kumar, A., Brüggemann, N., Hohenegger, C., and Ilyina, T.: Resolved tropical cyclones trigger $CO_2$ uptake and phytoplankton bloom in an Earth system model simulation, PNAS, 120, e2506103 122, https://doi.org/10.1073/pnas.2506103122, 2025.

Shu, Q., Wang, Q., Årthun, M., Wang, S., Song, Z., Zhang, M., and Qiao, F.: Arctic Ocean Amplification in a warming climate The rights exclusive Authors, reserved; licensee some in CMIP6 models, Science Advances, 8, eabn9755, 2022.

Takeuchi, M., Doubell, M. J., Jackson, G. A., Yukawa, M., Sagara, Y., and Yamazaki, H.: Turbulence mediates marine aggregate formation and destruction in the upper ocean, Scientific Reports, 9, 16 280, https://doi.org/10.1038/s41598-019-52470-5, 2019.

Visser, A. W.: Sequestration by the biological carbon pump: Do we really know what we are talking about?, Limnology and Oceanography Letters, https://doi.org/10.1002/lol2.70053, 2025.

Weber, T., Cram, J. A., Leung, S. W., DeVries, T., and Deutsch, C.: Deep ocean nutrients imply large latitudinal variation in particle transfer efficiency, PNAS, 113, 8606–8611, 2016.

Wilson, J. D., Andrews, O., Katavouta, A., de Melo Viríssimo, F., Death, R. M., Adloff, M., Baker, C. A., Blackledge, B., Goldsworth, F. W., Kennedy-Asser, A. T., Liu, Q., Sieradzan, K. R., Vosper, E., and Ying, R.: The biological carbon pump in CMIP6 models: 21st century trends and uncertainties, PNAS, 119, e2204369 119, https://doi.org/10.1073/pnas.2204369119, 2022.

---

## Author Comment (AC2)

**1 Responses to Reviewer II**

Dear Reviewer,

many thanks for you valuable comments on our manuscript. In the document below, please find

⇒ in blue your initial review comment

⇒ in darkcyan our reply, with potential indicated  → modifications/additions in the manuscript text

Kind regards,

     Joeran Maerz on behalf of the co-authors

**1.1 General comment**

Review of "Marine particles and their remineralization buffer future ocean biogeochemistry response to climate warming" by Maerz et al.

The manuscript by Maerz et al. provides an important synthesis of information related to particle formation, sinking, and remineralization processes in ocean biogeochemical models. The authors systematically compare a simple model version -referred to as "CMIP6"- with one that includes more complex particle-sinking and remineralization processes, "M4AGO", for both historical and future periods. They offer a detailed analysis of projected changes and assess the impact of using a more complex representation of particle sinking and remineralization.

The authors document the effect of representing marine particles and their remineralization in a more complex way (e.g., temperature- and oxygen-dependent remineralization, the effect of particle microstructure on sinking speed, such as the influence of ballast minerals) on climate projections. They clearly show that two regions most affected in terms of transfer efficiency are oxygen-deficient zones and the Arctic Ocean. These findings are highlighted in the results from the more complex model, "M4AGO" simulations. Authors also compare future changes (2070–2099) with historical periods (1985–2014) using two model versions: the simpler 'CMIP6' and the more complex 'M4AGO.' They demonstrate that marine particles play a role in buffering the future ocean biogeochemistry response to climate warming, especially in

the tropical and subtropical regions. They detail how a simple representation of marine particles in a climate model could alter projections of future p-ratios in these regions.

Overall, the manuscript is well written, well organized, and highly informative for the biogeochemical modeling community. It makes a significant contribution to ongoing research on the biological carbon pump using Earth system models. However, given the frequent references to Maerz et al. (2020) and Mauritsen et al. (2019) and the length of the manuscript, the Methods/Conclusion sections could benefit from adjustments to clarify the comparisons for readers.

$\Rightarrow$ Many thanks for the generally positive review of our manuscript. We will address the comments in the following.

**1.2 Specific comments**

My specific suggestions and comments are listed below:

Line 117: I suggest adding a small table or a simple illustration highlighting the differences between the CMIP6 version and M4AGO. As it is, the reader needs to refer back to Mauritsen et al. (2019) and Maerz et al. (2020) to fully understand the setup. A summary of the key differences would make the comparison easier to follow in the subsequent sections. A similar addition could be made for Section 2.2.

$\Rightarrow$ We will provide a brief table (see Tab. 1) with key-different parameterizations in the revised manuscript. However, we will try keeping Section 2 as short as possible, since we believe that Maerz et al. (2020) documented $M^4AGO$ and differences to the Martin curve approach extensively.

Table 1: Brief model differences between the CMIP6 and the M⁴AGO version. $[O_2]$ represents the oxygen concentration, $K_{O2} = 10\,\mu mol\,L^{-1}$ the half-saturation constant for oxygen limitation of aerobic remineralization and $T$ the local water temperature. M⁴AGO schematics taken from Maerz et al. (2020).

| | CMIP6 | M⁴AGO |
|---|---|---|
| **Sinking velocity** | | |
| POM $(m\,d^{-1})$ |  |  |
| Opal $(m\,d^{-1})$ | 30 | |
| CaCO$_3$ $(m\,d^{-1})$ | 30 | |
| Dust $(m\,d^{-1})$ | $\approx 0.05$ | |
| POM remineralization rate $(d^{-1})$ | $0.026 \cdot \frac{[O_2]}{[O_2+K_{O2}]}$ | $0.120 \cdot \frac{[O_2]}{[O_2+K_{O2}]} \cdot 2.1^{\frac{T-10}{10}}$ |
| Opal dissolution rate $(d^{-1})$ | $0.01 \cdot (0.1(T+3))$ | $0.023 \cdot 2.6^{\frac{T-10}{10}}$ |

Line 185: These kinds of metrics are difficult to standardize. In the biogeochemical modeling community, different metrics are used for similar analyses, but they represent different concepts, such as the f-ratio, e-ratio, p-ratio, and s-ratio. In this manuscript, the p-ratio is chosen to represent export efficiency, defined as the ratio of export flux to NPP. Since you frequently cite Laufkötter et al. (2016), it might be less confusing for readers referencing the same literature if you adopt consistent notation and explicitly cite that paper when introducing the metric. In Laufkötter et al. (2016), the p-ratio refers to the ratio of total POC to NPP, while the e-ratio is defined as export efficiency, the ratio of export flux to NPP. I am aware that p-ratio is also used in Maerz et al. (2020); I just wanted to raise this point for clarity, in case authors wish to change.

⇒ The reviewers comment let us to careful review available literature. We very much appreciate the aim for clarity and using common definitions among the OBGC community. However, we believe that the nomenclature in Laufkötter et al. (2016) is rather the exception to the norm in terms of nomenclature - re-defining some ratios defined/precisioned earlier in the literature. I.e. the $p$-ratio and $pe$-ratio was earlier defined by Brix et al. (2004), Dunne et al. (2005) and Brix et al. (2006) (while admittedly, we haven't cited these sources in Maerz et al. (2020) and should have used $pe$-ratio as nomenclature instead, which we will do when revising the manuscript). In the present manuscript, we thus refer to the original literature (Dunne et al., 2005; Brix et al., 2006) and briefly annotate the equivalent in Laufkötter et al. (2016) - their $e$-ratio that by definition of Laws et al. (2000) also encompasses dissolved organic matter fluxes and non-gravitational POM fluxes. We apologize to Dunne, Brix and co-authors to not having citet them properly initially in Maerz et al. (2020). We will now do so in the present manuscript.

Line 204: 'Higher remineralization':

- I noticed that the comparison of remineralization rates between the two models is not shown in any of the figures presented in the manuscript. Could this be added as a supplementary figure? Adding a figure would help confirm whether the observed differences are indeed due to higher remineralization.

⇒ We initially neglected to show the remineralization, since it was discussed in Maerz et al. 2020 (see their Fig. 9). Acknowledging the wish for a more comprehensive manuscript also by reviewer I, we will

explicitly mention values now in Sec. 2.1 (Brief model description), provide the values in the table (see above) and will provide a figure in the appendix. See Fig. 1 that will replace former figure D1.

[Figure]

Figure 1: Climatological year mean for the historical period and their changes in the future period in M$^4$AGO for a,b) sinking velocity; c,d) $Q_{10}$-dependent remineralization rate; e,f) remineralization length scales of POM. For comparison, the CMIP6 version features a globally constant sinking velocity of $3.5\,\mathrm{m\,d^{-1}}$ between $0\,\mathrm{m}$ to $100\,\mathrm{m}$ depth, a remineralization rate of $0.026\,\mathrm{d^{-1}}$ (times oxygen limitation) and thus a RLS(POC)$\approx 135\,\mathrm{m}$ at export depth (assuming no oxygen limitation here for simplicity).

- When I checked the sinking speed from the standard model, it appears to be $3.5\,\mathrm{m\,d^{-1}}$ at the top $100\,\mathrm{m}$. In contrast, in M4AGO, the concentration-weighted mean sinking velocity seems to be higher in the subtropics (Figure D1). Typically, one would expect higher nutrient export from the euphotic zone in a shorter time under such conditions. However, as stated, remineralization in the M4AGO case is significantly higher. Could you clarify how this balance between sinking speed and remineralization impacts nutrient export in models?

$\Rightarrow$ Indeed, instead of focusing only on sinking velocity, the remineralization length scales need to be considered when comparing two different sinking schemes in the same/similar circulation field to better understand loss of nutrients due to sinking and remineralization of POM to the mesopelagic. As Fig. 10 b shows only the zonal average of RLS, we will provide an additional subplot in Fig. D1 for the remineralization length scales for M$^4$AGO at export depth (and provide the globally constant value for the CMIP6 case, $\approx 136$ m - here under the assumption of well oxygenated waters), see Fig.1. Eventually, this translates into the pe-ratio as a measure of nutrient loss to the mesopelagic (shown in Fig. 3 of the manuscript). Thanks for hinting at that point to make it more explicit in the manuscript.

Regarding your decision to adopt temperature-dependent remineralization with Q10 factors, what was the motivation behind this choice? Would you expect that the results would change a lot depending on your Q10 choice?

$\Rightarrow$ During the development of M$^4$AGO, we reviewed the available literature and found a suggested range for an optimal POM remineralization $Q_{10}$ factor for the ocean spanning between 1.5 to 2.01 (Laufkötter et al., 2017) and $2.5 \pm 0.2$ (DeVries and Weber, 2017). In a detailed study on microbial remineralization dynamics of marine particles, Mislan et al. (2014) applied a $Q_{10} = 2.0$ based on an extensive physiological meta study by Dell et al. (2011). In the development/tuning process for M$^4$AGO published in Maerz et al. (2020), we saw some effects on the regional transfer efficiency values, when varying the $Q_{10}$ factor, but not on the overall global pattern of transfer efficiency. Further, the equatorial and subtropical gyre phosphate concentrations posed another constrain on varying the remineralization rate and $Q_{10}$ factor. For some sensitivity of the phosphate concentration and transfer efficiency on changing remineralization length scales, see Maerz et al. (2020), Fig. 15 - particularly by fixing $d_f$, which affects the sinking velocity as counterpart of remineralization, but also changing the frustule size, which also affects sinking velocity in silicifier-dominated regions. Eventually, we settled on the compromise between the three studies and chose $Q_{10} = 2.1$. While varying the $Q_{10}$ relatively easily let's investigate the effect on the remineralization rate (just graphically), its feedback on sinking velocity in M$^4$AGO is more difficult to assess. It would thus deserve an in-depth study which is outside the scope of manuscript, while it certainly could be a valuable study on its own.

**Line 208:** The statement about "increasing stratification, weaker mixing, and less recovery of exported nutrients" is compelling. However, it would be helpful to back this up with evidence showing the relationship between increased temperature or increased stratification in your model results. Including figures in the appendix would strengthen the argument and make the reasoning easier to follow.

⇒ Thanks for this recommendation. We will include a figure showing the climatological annual maximum mixed layer depth (MLD) of daily maximum values as measure for the winter mixed layer and the potential to recover nutrients from deeper ocean layers. Further, we will show the mean maximum vertical stratification (represented by the squared buoyancy frequency $N^2$) as a measure on how strong the potentially nutrient-deprived mixed layer is separated from the below, typically more nutrient-rich ocean layers. For both measures, we also show the future changes (see Fig.2, which we will include in the manuscript). For completeness for the discussion, we below also show the monthly changes of these measures which we do not intend to populate further into a revised manuscript version (see Fig. 3 and 4, CMIP6 simulation shows similar pattern, not shown).

**M⁴AGO climatological mean of year maximum MLD for the historical period and Δ max MLD = projection - historical period**

[Figure]

**M⁴AGO climatological year mean of monthly vertical maximum mean $N^2$ (0-500 m) for the historical period and $\Delta N^2$ = projection - historical period**

[Figure]

Figure 2: M⁴AGO climatological maximum mixed layer depth in the historical period and its changes in the future projection. Below: Climatological mean of vertical maximum of monthly mean squared buoyancy frequency, $N^2$, of the upper 500 m in the historical period and changes in the future period.

**M⁴AGO climatological mean of monthly maximum MLD for the historical period and Δ max MLD = projection - historical period**

[Figure]

Figure 3: Monthly resolved maximum mixed layer depth for the M⁴AGO simulation.

**M⁴AGO climatological mean of vertical maximum $N^2$ in top 500 m water column for the historical period and $\Delta N^2$ = projection - historical period**

[Figure]

Figure 4: Climatological mean of vertical maximum of $N^2$ in the upper 500 m and future changes globally - here for the M⁴AGO simulation.

**Line 251:** When I read Equation 2, the transfer efficiency appears to be independent of NPP. The primary driver of its change is the balance between sinking speed and remineralization. The manuscript states that adding POM increases the buoyancy of marine particles, thereby decreasing sinking velocity. Would it be more effective to integrate this explanation with the discussion on changes in particle properties in Section 3.4? While Appendix C also conveys this message, readers must carefully analyze the notations and navigate a rather crowded figure to understand it fully. Simplifying or consolidating these points in the main text could improve clarity.

$\Rightarrow$ We will consolidate these points in Sec. 3.4, when revising the manuscript. In Sec. 3.4, we will extend and clarify the part, where we address the high latitude changes in particle properties. In the context of line 251, we will thus reference to Sec. 3.4, instead of formerly Appendix C, while we believe that the short reference to particle buoyancy aids in grasping the reason for changes in transfer efficiency in the future Arctic Ocean. In addition, by providing a summary graphics, we belive to further summarize and clarify this behavior.

**Line 260:** Can you clarify how a reader can see that the Weddell and Ross seas from Figure 4c?

$\Rightarrow$ Any parts of the Weddell and Ross sea $\geq 1000$ m are encompassed in the transfer efficiency value for the antarctic zone. For detailed values of these regions, refer to Fig. 4a. While addressing this point, we noticed, though, that we should clarify that regions with depths smaller than 1000 m render the transfer efficiency metric as non-applicable, which is why it is set to NaN. We clarify this in the figures caption text. We add: $\rightarrow$ Ocean regions like shelf seas with water depths represented smaller than 1000 m are neglected and are displayed in white.

**Line 505-560:** The manuscript could benefit from a summary figure that highlights all key changes documented across the results sections (e.g., responses of Arctic, subtropical, and tropical regions). Adding such a figure would be beneficial, given the large amount of information presented in the paper, as it could help a broader audience beyond just biogeochemical modelers. A concise visual summary would make it easier for readers to understand and engage with the key findings.

$\Rightarrow$ Thanks for this consideration. It was also suggested by reviewer I and we aim at following the advice by including a summary figure - a radar plot. See comments to reviewer I.

**2 Typos:**

Figure 4 caption: Typo "standrad" - should be "standard".

⇒ Thanks. Typo will be corrected when revising the manuscript.

Line 497: Typo in "mesopelgic" - should be "mesopelagic".

⇒ Thanks. Typo will be corrected when revising the manuscript.

**References**

Armstrong, R. A., Lee, C., Hedges, J. I., Honjo, S., and Wakeham, S. G.: A new, mechanistic model for organic carbon fluxes in the ocean based on the quantitative association of POC with ballast minerals, Deep-Sea Research II, 49, 219–236, 2002.

Brix, H., Gruber, N., and Keeling, C. D.: Interannual variability of the upper ocean carbon cycle at station ALOHA near Hawaii, Global Biogeochemical Cycles, 18, GB4019, 2004.

Brix, H., Gruber, N., Karl, D. M., and Bates, N. R.: On the relationships between primary, net community, and export production in subtropical gyres, Deep-Sea Research II, 53, 698–717, 2006.

Dell, A. I., Pawar, S., and Savage, V. M.: Systematic variation in the temperature dependence of physiological and ecological traits, PNAS, 108, 10 591–10 596, 2011.

DeVries, T. and Weber, T.: The export and fate of organic matter in the ocean: New constraints from combining satellite and oceanographic tracer observations, Global Biogeochemical Cycles, 31, 535–555, 2017.

Dunne, J. P., Armstrong, R. A., Gnanadesikan, A., and Sarmiento, J. L.: Empirical and mechanistic models for the particle export ratio, Global Biogeochemical Cycles, 19, GB4026, https://doi.org/10.1029/2004GB002390, 2005.

Iversen, M. H. and Ploug, H.: Ballast minerals and the sinking carbon flux in the ocean: carbon-specific respiration rates and sinking velocity of marine snow aggregates, Biogeosciences, 7, 2613–2624, 2010.

Laufkötter, C., Vogt, M., Gruber, N., Aumont, O., Bopp, L., Doney, S. C., Dunne, J. P., Hauck, J., John, J. G., Lima, I. D., Séférian, R., and Völker, C.: Projected decreases in future marine export production: the role of the carbon flux through the upper ocean ecosystem, Biogeosciences, 13, 4023–4047, https://doi.org/10.5194/bg-13-4023-2016, 2016.

Laufkötter, C., John, J. G., Stock, C. A., and Dunne, J. P.: Temperature and oxygen dependence of the remineralization of organic matter, Global Biogeochemical Cycles, 31, 1038–1050, https://doi.org/10.1002/2017GB005643, 2017.

Laws, E. A., Falkowski, P. G., Smith Jr., W. O., Ducklow, H., and McCarthy, J. J.: Temperature effects on export production in the open ocean, Global Biogeochemical Cycles, 14, 1231–1246, 2000.

Maerz, J., Six, K. D., Stemmler, I., Ahmerkamp, S., and Ilyina, T.: Microstructure and composition of marine aggregates as co-determinants for vertical particulate organic matter transfer in the global ocean, Biogeosciences, 17, 1765–1803, 2020.

Mauritsen, T., Bader, J., Becker, T., Behrens, J., Bittner, M., Brokopf, R., Brovkin, V., Claussen, M., Crueger, T., Esch, M., Fast, I., Fiedler, S., Fläschner, D., Gayler, V., Giorgetta, M., Goll, D. S., Haak, H., Hagemann, S., Hedemann, C., Hohenegger, C., Ilyina, T., Jahns, T., Jimenéz-de-la Cuesta, D., Jungclaus, J., Kleinen, T., Kloster, S., Kracher, D., Kinne, S., Kleberg, D., Lasslop, G., Kornblueh, L., Marotzke, J., Matei, D., Meraner, K., Mikolajewicz, U., Modali, K., Möbis, B., Müller, W. A., Nabel, J. E. M. S., Nam, C. C. W., Notz, D., Nyawira, S.-S., Paulsen, H., Peters, K., Pincus, R., Pohlmann, H., Pongratz, J., Popp, M., Raddatz, T. J., Rast, S., Redler, R., Reick, C. H., Rohrschneider, T., Schemann, V., Schmidt, H., Schnur, R., Schulzweida, U., Six, K. D., Stein, L., Stemmler, I., Stevens, B., von Storch, J.-S., Tian, F., Voigt, A., Vrese, P., Wieners, K.-H., Wilkenskjeld, S., Winkler, A., and Roeckner, E.: Developments in the MPI-M Earth System Model version 1.2 (MPI-ESM1.2) and Its Response to Increasing $CO_2$, Journal of Advances in Modeling Earth Systems, 11, 1–41, https://doi.org/10.1029/2018MS001400, 2019.

Mislan, K. A. S., Stock, C. A., Dunne, J. P., and Sarmiento, J.: Group behaviour among model bacteria influences particulate carbon mineralization depths, Journal of Marine Research, 72, 183–218, 2014.